



**Title:** Fluorescent Bioaerosol Particle, Molecular Tracer, and Fungal Spore Concentrations during Dry
2       and Rainy Periods in a Semi-Arid Forest
**Authors:** Marie Ila GOSSELIN[1,2], Chathurika M Rathnayake[3], Ian Crawford[4], Christopher Pöhlker[2],
6       Janine Fröhlich-Nowoisky[2], Beatrice Schmer[2], Viviane R. Després[5], Guenter Engling[6], Martin
7       Gallagher[4], Elizabeth Stone[3], Ulrich Pöschl[2], and J. Alex Huffman[1*]
[1]Department of Chemistry and Biochemistry, University of Denver, Denver, Colorado, USA
[2]Max Planck Institute for Chemistry, Multiphase Chemistry and Biogeochemistry Departments, Mainz,
Germany
[3]Department of Chemistry, University of Iowa, Iowa City, IA 52246, USA
[4]Centre for Atmospheric Science, SEAES, University of Manchester, Manchester, UK
[5]Institute of General Botany, Johannes Gutenberg University, Mainz, Germany
[6]Division of Atmospheric Sciences, Desert Research Institute, Reno, NV, USA
* Correspondence to: alex.huffman@DU.edu



**Abstract:**
Bioaerosols pose risks to human health and agriculture and may influence the evolution of mixed-phase
clouds and the hydrological cycle on local and regional scales. The availability and reliability of methods
and data on the abundance and properties of atmospheric bioaerosols, however, are rather limited. Here
we analyze and compare data from different real-time Ultraviolet Laser/Light Induced Fluorescence (UV-
LIF) instruments with results from a culture-based spore sampler and offline molecular tracers for
airborne fungal spores in a semi-arid forest in the Southern Rocky Mountains of Colorado. Commercial
UV-APS (Ultraviolet Aerodynamic Particle Sizer) and WIBS-3 (Wideband Integrated Bioaerosol Sensor,
Version 3) instruments with different excitation and emission wavelengths were utilized to measure
fluorescent aerosol particles (FAP) during both dry weather conditions and periods heavily influenced by
rain. Seven molecular tracers of bioaerosols were quantified by analysis of total suspended particle (TSP)
high-volume filter samples using High Performance Anion Exchange Chromatography with Pulsed
Amperometric Detection (HPAEC-PAD). From the same measurement campaign Huffman et al. (2013)
previously reported dramatic increases in total and fluorescence particle concentrations during and
immediately after rainfall and also showed a strong relationship between the concentrations of FAP and
ice nuclei (Huffman et al., 2013; Prenni et al., 2013). Here we investigate molecular tracers and show that
during rainy periods the atmospheric concentrations of arabitol ($35.2 \pm 10.5$ ng m$^{-3}$) and mannitol ($44.9 \pm$
$13.8$ ng m$^{-3}$) were 3-4 times higher than during dry periods. During and after rain the correlations between
FAP and tracer mass concentrations were also significantly improved. Fungal spore number
concentrations on the order of $10^4$ m$^{-3}$, accounting for 2-4% of TSP mass during dry periods and 17-23%
during rainy periods, were obtained from scaling the tracer measurements and from multiple analysis
methods applied to the UV-LIF data. Endotoxin concentrations were also enhanced during rainy periods,
but showed no correlation with FAP concentrations. Average mass concentrations of erythritol,
levoglucosan, glucose, and $(1\rightarrow3)$-β-D-glucan in TSP samples are reported separately for dry and rainy
weather conditions. Overall, the results indicate that UV-LIF measurements can be used to infer fungal
spore concentrations, but substantial development of instrumental and data analysis methods seems
required for improved quantification.





## 1. Introduction

Primary biological aerosols particles (PBAP) are of keen interest within the scientific community,
partially because methods for their quantification and characterization are advancing rapidly (Huffman
and Santarpia, 2016; Sodeau and O'Connor, 2016). The term PBAP, or equivalently bioaerosol, generally
comprises several classes of airborne biological particles including viruses, bacteria, fungal spores, pollen
and their fragments (Després et al., 2012; Fröhlich-Nowoisky, 2016). Fungal spores are of particular
atmospheric interest because they can cause a variety of deleterious health effects in humans, animals,
and agriculture, and it has been shown that they can represent a significant fraction of total organic
aerosol emissions (Deguillaume et al., 2008; Gilardoni et al., 2011; Madelin, 1994), especially in tropical
regions (Elbert et al., 2007; Huffman et al., 2012; Pöschl et al., 2010; Zhang et al., 2010). Current
estimates of the atmospheric concentration of fungal spores range from $10^0$ to more than $10^4$ m$^{-3}$
(Frankland and Gregory, 1973; Gregory and Sreeramulu, 1958; Heald and Spracklen, 2009; Hummel et
al., 2015; Sesartic and Dallafior, 2011). Fungal spores may also impact the hydrological cycle as giant
cloud condensation nuclei (GCCN) or as ice nuclei (IN) (Haga et al., 2013; Morris et al., 2013; Sesartic et
al., 2013). Additionally, several classes of bioaerosols and their constituent components, such (1→3)-β-
D-glucan and endotoxins, have been implicated in respiratory distress and allergies (Burger, 1990;
Douwes et al., 2003; Laumbach and Kipen, 2005; Pöschl and Shiraiwa, 2015). For example, asthma and
allergies have shown notable increases during thunderstorms due to elevated bioaerosol concentrations
(Taylor and Jonsson, 2004) especially when attributed to fungal spores (Allitt, 2000; Dales et al., 2003).
Molecular tracers have long been utilized as a means of aerosol source tracking (Schauer et al.,
1996; Simoneit and Mazurek, 1989; Simoneit et al., 2004). In recent years, analysis of molecular tracers
has been utilized for the quantification of PBAP in atmospheric samples and has been compared with
results from microscopy (Bauer et al., 2008a) and culture samples (Chow et al., 2015b; Womiloju et al.,
2003). Three organic molecules have been predominately utilized as unique tracers of fungal spores:
ergosterol, mannitol, and arabitol. The majority of atmospherically relevant fungal spores are released by
active wet discharge processes common in *Ascomycota* and *Basidiomycota*, meaning that the fungal
organism actively ejects spores at a time most advantageous for the spore dispersal and germination
processes, often when relative humidity (RH) is high (Ingold, 1971). While there are several mechanisms
of active spore emission (e.g. Buller's drop (Buller, 1909) and osmotic pressure canons (Ingold, 1971)),
they each involve the secretion of fluid containing hygroscopic compounds, such as arabitol, mannitol,
potassium, chloride, and other solutes (Elbert et al., 2007), released near the site of spore growth. When
the spores are ejected, some of the fluid adheres to the spores and becomes aerosolized. Several of these
secreted compounds are thought to enter the atmosphere linked uniquely with spore emission processes,
and so these tracers have been used to estimate atmospheric concentrations of fungal spores. Arabitol and
mannitol are both sugar alcohols (polyols) that serve as energy stores for the spore (Feofilova, 2001).
Arabitol is unique to fungal spores and lichen, while mannitol is present in fungal spores, lichen, algae,
and higher plants (Lewis and Smith, 1967). Ergosterol is found within the cell membranes of fungal
spores (Weete, 1973) and can be used as an ambient fungal spore trace (Di Filippo et al., 2013; Miller and
Young, 1997). Comparing the seasonal trends of arabitol and mannitol with ergosterol, Burshtein et al.
(2011) showed positive correlations between arabitol or mannitol and ergosterol only in the spring and
autumn suggesting that the source of these polyols is unlikely to be solely fungal in origin or that the
amount of each compound emitted varies considerably between species type and season. While ergosterol
has been directly linked to fungal spores in the air, ergosterol is prone to photochemical degradation and
is difficult to analyze and quantify directly. Quantification of ergosterol typically requires chemical
derivatization by silylation before analysis via gas chromatography (Axelsson et al., 1995; Burshtein et
al., 2011; Lau et al., 2006). In contrast, analysis of sugar alcohols by ion chromatography involves fewer
steps and has been successfully applied to monitor seasonal variations of atmospheric aerosol
concentration at a number of sites (Bauer et al., 2008a; Caseiro et al., 2007; Yang et al., 2012; Yttri et al.,
2011a; Zhang et al., 2010; Zhang et al., 2015) including pg m$^{-3}$ levels in the Antarctic (Barbaro et al.,
2015). By measuring spore count and tracer concentration in parallel at one urban and two suburban sites

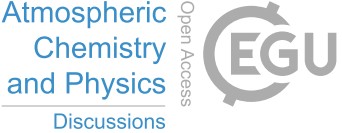

in Vienna, Austria, Bauer et al. (2008a) estimated the amount of each tracer per fungal spore emitted.
Potassium ions have also been linked to emission of biogenic aerosol (Pöhlker et al., 2012b) and are co-
emitted with fungal spores, however, application of potassium as a fungal tracer is uncommon because it
is predominantly associated with biomass burning (Andreae and Crutzen, 1997). Additionally, (1→3)-β-
D-glucan (fungal spores and pollen) and endotoxins (gram-negative bacteria) have also been widely used
to measure other bioaerosols (Andreae and Crutzen, 1997; Cheng et al., 2012; Stone and Clarke, 1992).

The direct detection of PBAP has historically been limited to analysis techniques that require
culturing or microscopy of the samples. These systems are time-consuming, costly, and often
substantially undercount biological particles by an order of magnitude or more (Gonçalves et al., 2010;
Pyrri and Kapsanaki-Gotsi, 2007). The sampling methods associated with these measurements also offer
relatively low time resolution and low particle size resolution. Recently, techniques utilizing ultraviolet
laser/light-induced fluorescence (UV-LIF) for the real-time detection of PBAP have been developed and
are being utilized by the atmospheric community for bioaerosol detection. Thus far, the most widely
applied LIF instruments for ambient PBAP detection have been the Ultraviolet Aerosol Particle Sizer
(UV-APS; TSI Inc. Model 3314, St. Paul, MN) and the Wideband Integrated Bioaerosol Sensor (WIBS;
University of Hertfordshire, Hertfordshire, UK, now licensed to Droplet Measurement Technologies,
Boulder, CO, USA). Both of these commercially available instruments can provide information in real-
time about particle size and fluorescence properties of supermicron atmospheric aerosols.
Characterization and co-deployment of these instruments over the past ten years has expanded the
knowledge base regarding how to analyze and utilize the information provided from these instruments
(Healy et al., 2014; Huffman et al., 2013; Pohlker et al., 2013; Pöhlker et al., 2012a), though the
interpretation of UV-LIF results from individual particles is complicated by interfering material that is not
biological in nature (Gabey et al., 2010; Huffman et al., 2012; Lee et al., 2010; Saari et al., 2013; Toprak
and Schnaiter, 2013).

Here we present analysis of atmospheric concentrations of arabitol and mannitol in relation to
results from real-time, ambient particle measurements reported by UV-APS and WIBS. We interrogate
these relationships as they pertain to rain conditions (rainfall and RH) that have previously been shown to
increase fluorescent aerosol concentration (Crawford et al., 2014; Huffman et al., 2013; Prenni et al.,
2013; Schumacher et al., 2013). Active wet discharge of ascospores and basidiospores has frequently
been reported to correspond with increased RH (Elbert et al., 2007), and fungal spore concentration has
also been shown to increase after rain events (e.g. Jones and Harrison, 2004). Here we estimate airborne
fungal concentrations in a semi-arid forest environment utilizing a combination of real-time fluorescence
methods, molecular fungal tracer methods, and direct-to-agar sampling and culturing as parallel
surrogates for spore analysis. This study represents the first ambient comparison of real-time aerosol UV-
LIF instruments with results from molecular tracers or culturing.
**2. Methods**
**2.1 Sampling site**
Atmospheric sampling was conducted as a part of the BEACHON-RoMBAS (Bio-hydro-
atmosphere interactions of Energy, Aerosols, Carbon, H2O, Organics, and Nitrogen – Rocky Mountain
Biogenic Aerosol Study) field campaign conducted at the Manitou Experimental Forest Observatory
(MEFO) located 48 km northwest of Colorado Springs, Colorado (2370 m elevation, 39° 06' 0" N, 105°
5' 03" W) (Ortega et al., 2014). The site is located in the central Rocky Mountains and is representative of
semi-arid montane pine forested regions of North America. During BEACHON-RoMBAS, a large team
of international researchers conducted an intensive set of measurements from 20 July to 23 August 2011.
A summary of results from the campaign are published in the BEACHON campaign special issue of



Atmospheric Chemistry and Physics[1]. All the data used in this study were gathered from instruments and
sensors located within a <100 m radius (e.g. Fig. 1).

**2.2 Online fluorescent instruments**
A UV-APS and WIBS-3 (Model 3; University of Hertfordshire) were operated continuously as a
part of the study, and particle data were integrated to five-minute averages before analysis. The UV-APS
was operated under procedures defined in previous studies (Huffman et al., 2013; Schumacher et al.,
2013). A total suspended particle (TSP) inlet head ~5.5 m above ground, mounted above the roof of a
climate-controlled, metal trailer, was used to sample aerosol directed towards the UV-APS. Bends and
horizontal stretches in the 0.75 inch tubing were minimized to reduce losses of large particles (Huffman et
al., 2013). The UV-APS detects particles between 0.5-20 μm and records aerodynamic particle diameter
and integrated total fluorescence (420-575 nm) after pulsed excitation by a 355 nm laser (Hairston et al.,
1997). Both UV-APS and WIBS instruments report information about particle number concentration, but
it is instructive here to show results in particle mass for comparison between all techniques. Total particle
number size distributions (irrespective of fluorescence properties) obtained from the UV-APS were
converted to mass distributions using unit particle mass density of as a first approximation for all direct
comparisons with tracer mass and unless otherwise stated. Total particle concentration values (in μg m$^{-3}$)
were obtained for each five-minute period by integrating over the size range 0.5 – 15 μm, and these mass
concentration values were averaged over the length of the filter sampling periods. Uncertainty in mass
concentration values reported here is influenced by utilizing a single, estimated value for particle mass
density and because of slight dissimilarities between UV-APS and WIBS instruments in size binning at
particle sizes above 10 μm that dominate particle mass.

A WIBS-3 was used to continuously sample air at a site ~50 m away from the UV-APS trailer
and 1.3 m above the ground. Briefly, the diameter of individual particles sampled by the WIBS is
estimated by the intensity of the elastic side-scatter from a continuous wave 635 nm diode laser and
analyzed by a Mie scattering model (Foot et al., 2008; Kaye et al., 2005). Particles that pass through the
diode laser activate two optically-filtered Xenon flash lamps. The first lamp excites the particle at 280 nm
and the second at 370 nm. Emission from the 280 nm excitation is filtered separately for two PMTs, one
which detects in a band at 310-400 nm and the other in a band at 410-650 nm. These excitation and
emission wavelengths result in a total of three channels of detection: $\lambda_{ex}$ 280 nm, $\lambda_{em}$ 320 – 400 nm (FL 1
or Channel A); $\lambda_{ex}$ 280 nm, $\lambda_{em}$ 410 – 650 nm (FL 2 or Channel B); and $\lambda_{ex}$ 370 nm, $\lambda_{em}$ 410– 650 nm (FL
3 or Channel C) (Foot et al., 2008; Gabey et al., 2010; Perring et al., 2015). Individual particles are
considered fluorescent here if they exceed fluorescent thresholds for any channel, as defined as the
average of a "forced trigger" baseline plus 3 standard deviations (σ) of the baseline measurement (Gabey
et al., 2010).

WIBS particle-type analysis is utilized to define types of particles that have specific spectral
patterns. As defined by Perring et al. (2015), the 3 different fluorescent channels (FL1, FL2, and FL3) can
be combined to produce 7 unique fluorescent categories. Observed fluorescence in channel FL1 alone, but
without any detectable fluorescence in Channel FL2 or FL3, categorizes a particle as type A. Similarly,
observed fluorescence in channels FL2 or FL3, but in no other channels, places a particle in the B or C
categories, respectively. Combinations of fluorescence in these channels, such as a particle that exhibits
fluorescence in both FL1 and FL2 categorizes a particle as type AB and so on for a possible seven particle
types as summarized in Figure S1.

As a separate tool for particle categorization, the University of Manchester has recently
developed and applied a hierarchical agglomerative cluster analysis tool for WIBS data, which they have

---

[1]http://www.atmos-chem-phys.net/special_issue247.html

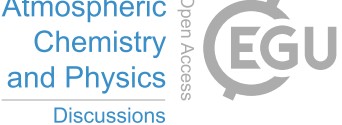

applied to the BEACHON-RoMBAS campaign (Crawford et al., 2014; Crawford et al., 2015; Robinson et
al., 2013). Here we utilize clusters derived from WIBS-3 data as described by Crawford et al. (2015).
Cluster data presented here was analyzed with the Open Source Python package FastCluster (Mullner,
2013). Briefly, hierarchical agglomerative cluster analysis was applied to the entire data set and each
fluorescent particle was uniquely clustered into one of 4 groups. Cluster 1, assigned by Crawford et al.
(2015) as fungal spores, displayed a 1.5-2 µm mode and a daily peak in the early morning that paralleled
relative humidity (Schumacher et al., 2013). Clusters 2, 3, and 4 have strong, positive correlations with
rainfall and exhibit size modes that peak at <1.2 µm and were initially described by Crawford et al. as
bacterial particles. Here we have summed Clusters 2-4 to a single group referred to as $Cl_{Bact}$, for
simplicity when comparing with molecular tracers.
The WIBS-3 utilized here has since been updated to the WIBS-4 (Univ. Hertfordshire, UK) and
WIBS-4A (Droplet Measurement Technologies, Boulder, Colorado). One important difference between
the models is that the WIBS-3 exhibits comparatively weak FL1 and FL2 signals with respect to the more
updated models, and is thus more influenced by FL3. This results in a different break-down of channel
intensity between instrument models, as will be discussed later.
**2.3 High volume sampler**
Total suspended particle samples were collected for molecular tracer and molecular genetic
analyses using a high volume sampler (Digitel DHA-80) drawing 1000 L min$^{-1}$ through 15 cm glass fiber
filters (Macherey-Nagel GmbH, Type MN 85/90, 406015, Düren, Germany) over a variety of sampling
times ranging from 4-48 h (supplemental Table S1). The sampler was located <50 m from each of the
UV-LIF instruments described here, approximately between the WIBS-3 and UV-APS. Prior to sampling
all filters were baked at 500 ºC for 12 h to remove DNA and organic contaminants. Samples were stored
in pre-baked aluminum bags after sampling at -20 ºC for 1-30 days and then at -80 ºC after overnight,
international transport cooled on dry ice. Due to the low vapor pressure of the molecular tracers analyzed
loss due to volatilization is considered unlikely (Zhang et al., 2010). 36 samples were collected during the
study, in addition to handling field blanks and operational field blanks. Handling blanks were acquired by
placing a filter into the sampler and immediately removing, without turning on the air flow control.
Operational blanks were placed into the sampler and exposed to 10 seconds of air flow.
**2.4 Slit Sampler**
A direct-to-agar slit sampler (Microbiological Air Sampler STA-203, New Brunswick Scientific
Co, Inc., Edison, NJ) was used to collect culturable airborne fungal spores. The sampler was placed ~2 m
above ground on a wooden support surface with 5 cm x 5 cm holes to allow air flow both up and down
through the support structure. Sampled air was drawn over the 15 cm diameter sampling plate filled with
growth media at a flow rate of 28 L min$^{-1}$ for sampling periods of 20 to 40 min. Growth media (malt
extract medium) was mixed with antibacterial agents (40 units streptomycin, Sigma Aldrich; 20 units
ampicillin, Fisher Scientific) to suppress bacterial colony growth. Plates were prepared several weeks in
advance and stored in a refrigerator at ca. 4 ºC until used for sampling. Before each sampling period, all
surfaces of the samplers were sterilized by wiping with isopropyl alcohol. Handling and operational
blanks were collected to verify that no fungal colonies were being introduced by handling procedures. 14
air samples were collected over 20 days and immediately moved to an incubator (Amerex Instruments,
Incumax IC150R) set at 25 ºC for 3 days prior to counting fungal colonies formed. Each colony, present
as a growing dot on the agar surface, is assumed to have originated as one colony forming unit (CFU; i.e.
fungal spore) deposited onto the agar by impaction during sampling. The atmospheric concentration of
CFU per air volume was calculated using the sampler air flow. Further discussion of methods and initial
results from the slit sampler were published by Huffman et al. (2013).
**2.5 Offline filter analyses**



*2.5.1 Carbohydrate analysis*

Approximately 1/8 of each frozen filter was cut for carbohydrate analysis using a sterile
technique, meaning that scissors were cleaned and sterilized and cutting was performed in a positive-
pressure laminar flow hood. In order to precisely determine the fractional area of the filter to be analyzed,
filters were imaged from a fixed distance above using a camera and compared to a whole, intact filter.
Using ImageJ software (Rasband and ImageJ, 1997), the area of each filter slice showing particulate
matter (PM) deposit was referenced to a whole filter, and thereby the amount of each filter utilized could
be determined. This technique allowed for an estimate of the fraction of each sampled used for the
analysis, which corresponds to the fraction of PM mass deposited. The uncertainty on the filter area
fraction is estimated at 2%. The uncertainty was determined as the percent of variation in the area of the
filter edge (no PM deposit) as compared to the total filter area.
Carbohydrates were extracted from quartz filter samples and analyzed following the procedure
described by Rathnayake et al. (2016). A total of 36 samples were analyzed along with field and lab
blanks. All lab and field blanks fell below method detection limits. Extraction was performed by placing
the filter slice into a centrifuge tube that had been pre-rinsed with Nanopure™ water (resistance > 18.2
MΩ cm$^{-1}$; Barnstead EasyPure II, 7401). A volume of 8.0 mL of Nanopure™ water was added to the filter
in the centrifuge tube to extract water-soluble carbohydrates. Samples were then exposed to rotary
shaking for 10 min at 125 rpm, sonication for 30 min at 60 Hz (Branson 5510, Danbury, CT, US), and
rotary shaking for 10 min. After shaking, the extracted solutions were filtered through a 0.45 µm
polypropylene syringe filter (GE Healthcare, UK) to remove insoluble particles, including disintegrated
filter pieces. One 1.5 mL aliquot of each extracted solution was analyzed for carbohydrates within 24
hours of extraction. A duplicate 1.5 mL aliquot was stored in a freezer and analyzed, if necessary due to
lack of instrument response and invalid calibration check, within 7 days of extraction. Analysis of
carbohydrates was done using a High Performance Anion Exchange Chromatography System with Pulsed
Amperometric Detection (HPAEC-PAD, Dionex ICS 5000, Thermo Fisher, Sunnyvale, CA, USA).
Details of the instrument specifications and quality standards for carbohydrate determination are available
in Rathnayake et al. (2016). Calibration curves for mannitol, levoglucosan, glucose (Sigma-Aldrich),
arabitol and erythritol (Alfa Aesar) were generated with seven points each, ranging in aqueous
concentration from 0.005 ppm to 5 ppm. The method detection limits for mannitol, levoglucosan, glucose,
arabitol and erythritol were 2.3, 2.8, 1.6, 1.0 and 0.6 ppb, respectively. Method detection limits were
determined as 3σ of analyte concentrations recovered from seven spiked filter samples (Rathnayake et al.,
2016). All calibration curves were checked daily using a standard solution to ensure all concentration
values were within 10% of the known value. Failure to maintain a valid curve resulted in recalibration of
the instrument.

*2.5.2 DNA analysis*

Methods and initial results from DNA analysis from these high volume filters were published by
Huffman et al. (2013). Briefly, fungal diversity was determined by previously optimized methods for
DNA extraction, amplification, and sequence analysis of the internal transcribed spacer regions of
ribosomal genes from the high volume filter samples (Fröhlich-Nowoisky et al., 2012; Fröhlich-
Nowoisky et al., 2009). Upon sequence determination, fungal sequences were compared with known
sequences using the Basic Local Alignment Search Tool (BLAST) at the National Center for
Biotechnology (NCBI) and identified to the lowest taxonomic rank common to the top BLAST hits after
chimeric sequences had been removed. When sequences displayed >97% similarity, they were grouped
into operational taxonomic units (OTUs).

*2.5.3 Endotoxin and glucan analysis*

Sample preparation for quantification of endotoxin and (1→3)-β-D-glucan included extraction of
5 punches (0.5 cm$^2$ each) of the quartz filters with 5.0 mL of pyrogen-free water (Associates of Cape Cod
Inc., East Falmouth, MA, USA), utilizing an orbital shaker (300 rpm) at room temperature for 60 min,





followed by centrifuging for 15 min (1000 rpm). One-half mL of supernatant was submitted to a kinetic
chromogenic limulus amebocyte lysate (Chromo-LAL) endotoxin assay (Associates of Cape Cod Inc.,
East Falmouth, MA, USA) using a ELx808IU (BioTek Instrument Inc., Winooski, VT, USA) incubating
absorbance microplate reader. For (1→3)-β-D-glucan measurement, 0.5 mL of 3 N NaOH was added to
the remaining 4.5 mL of extract and the mixture was agitated for 60 min. Subsequently, the solution was
neutralized to pH 6–8 by addition of 0.75 mL of 2 N HCl. After centrifuging for 15 min (1→3)-β-D-
glucan concentration was determined in the supernatant using the Glucatell® LAL kinetic assay
(Associates of Cape Cod, Inc., East Falmouth, MA). The minimum detection limits (MDLs) and
reproducibility were 0.046 Endotoxin Units (EU) m$^{-3}$ and $\pm$ 6.4% for endotoxin and 0.029 ng m$^{-3}$ and $\pm$
4.2% for (1→3)-β-D-glucan, respectively. Laboratory and field blank samples were analyzed as well,
with lab blank values being below detection limits, while field blank values were used to subtract
background levels from sample data. More details about the bioassays can be found elsewhere (Chow et
al., 2015a).
**2.6 Meteorology and wetness sensors**
Meteorological data were recorded by a variety of sensors located at the site. Precipitation was recorded
by a laser optical disdrometer (PARticle SIze and VELocity "PARSIVEL" sensor; OTT Hydromet
GmbH, Kempton, Germany) and separately by a tipping bucket rain gauge. The disdrometer provides
precipitation occurrence, rate, and physical state (rain or hail) by measuring the magnitude and duration
of disruption to a continuous 780 nm laser that was located in a tree clearing (Fig. 1), while the tipping
bucket rain gauge measures a set amount of precipitation before tipping and triggering an electrical pulse.
A leaf wetness sensor (LWS; Decagon Devices, Inc., Pullman, WA), provided a measurement of
condensed moisture by measuring the voltage drop across a leaf surface to determine a proportional
amount of water on or near the sensor. Additional details of these measurements can be found in Huffman
et al. (2013) and Ortega et al. (2014).
**3. Results and Discussion**
**3.1 Categorization and characteristic differences of Dry and Rainy periods**
Increases in PBAP concentration have been frequently associated with rainfall (e.g. Bigg et al.,
2015; Faulwetter, 1917; Hirst and Stedman, 1963; Jones and Harrison, 2004; Madden, 1997). Fungal
polyols have also been reported to increase after rain and have been used as indicators of increased fungal
spore release (Liang et al., 2013; Lin and Li, 2000; Zhu et al., 2015). Recently it was shown that the
concentration of fluorescent aerosol particles (FAP) measured during BEACHON-RoMBAS increased
dramatically during and after periods of rain (Crawford et al., 2014; Huffman et al., 2013; Schumacher et
al., 2013) and that these particle were associated with high concentrations of ice nucleating particles that
could influence the formation and evolution of mixed-phase clouds (Huffman et al., 2013; Prenni et al.,
2013; Tobo et al., 2013). It was observed that a mode of smaller fluorescent particles (2-3 µm) appeared
during rain episodes, and several hours after rain ceased a second mode of slightly larger fluorescent
particle (4-6 µm) emerged, persisting for up to 12 h (Huffman et al., 2013). The first mode was
hypothesized to result from mechanical ejection of particles due to rain splash on soil and vegetated
surfaces, and the second mode was suggested as actively emitted fungal spores (Huffman et al., 2013).
While the UV-APS and WIBS each provide data at high enough time resolution to see subtle changes in
aerosol concentration, the temporal resolution of the chemical tracer analysis was limited to 4-48 h
periods defined by the collection time of the high volume sampler. To compare the measurement results
across the sampling platforms, UV-LIF measurements were averaged to the lower time resolution of the
filter sampler periods, and the periods were grouped into three broad categories: Rainy, Dry, and Other, as
will be defined below.
Time periods were wetness-categorized in two steps: first at 15 min resolution and then averaged
for each individual filter sample. During the first stage of categorization each 15 min period was
categorized into one of four groups: rain, post-rain, dry, or other. To categorize each filter period, an





algorithm was established utilizing UV-APS fluorescent particle fraction and accumulated rainfall. The
ratio of integrated number of fluorescent particles to total particles was used as a proxy for the increased
emission of biological particles. Figure 2a presents a time series of the size-resolved fluorescent particle
concentration, showing increases during rain periods in dark red. A relatively consistent diurnal cycle of
increased FAP concentration in the 2-4 μm range is apparent almost every afternoon, which corresponds
to near daily afternoon rainfall during approximately the first half of the measurement period.
Disdrometer and tipping bucket rainfall measurements were each normalized to unity and summed to
produce a more robust measure of rainfall rate, because it was observed that often only one of the two
systems would record a given light rain event. If a point was described by total rainfall accumulation
greater than 0.201 it was flagged as rain. A point was flagged as post-rain if it immediately followed a
rain period and also exhibited a fluorescent particle fraction greater than 0.08. The purpose of this
category was to reflect the observation that sustained, elevated concentrations of FAP persisted for many
hours even after the rain rate, RH, and leaf wetness returned to pre-rain values. The only measurement
that adequately reflected this scenario was of the fluorescent particles measured by UV-APS and WIBS
instruments. The post-rain flag was continued until the fluorescent particle fraction fell below 0.08 or if it
started to rain again (with calculated rain values greater than 0.201). Points were flagged as dry periods if
they exhibited rainfall accumulation and fluorescent particle fraction below the thresholds stated above.
Several periods were not easily categorized by this system and were considered in a fourth category as
other. This occurred when fluorescent particle fraction above the threshold value was observed with no
rainfall.
Once wetness categories were assigned by the algorithm at 15 min resolution, each high volume
filter sample was categorized by a similar nomenclature, but using only three categories. These were
defined as Dry, Rainy (combination of rain and post rain categories), or Other based on the relative time
fraction in each of the four original 15 min categories. For each sample, if the relative time fraction of a
given category exceeded 0.50 the sample was assigned to that category. Despite the effort to categorize
samples systematically, several sample periods (5 of 35) appeared mis-categorized by looking at FAP
concentration, rainfall, RH, and leaf wetness in more detail. In some circumstances, this was because light
rainfall produced observable increases in FAP, but without exceeding the rainfall threshold. Or in other
circumstances a period of rainfall occurred at the very end or just before the beginning of a sample, and so
the many-hour period was heavily influenced by aerosol triggered by a period of rain just outside of the
sample time window. As a result, several samples were manually re-categorized as described here.
Samples 20 and 21 (Table S1) were four-hour samples that displayed high relative humidity and rainfall,
thus samples were originally characterized as Rainy. This period was described by an extremely heavy
rain downpour (7.5 mm in 15 min), however, that seemingly placed the samples in a different regime of
rain-aerosol dynamics than the other Rainy samples and so these two samples were moved to the Other
category. Sample 23, originally Rainy, presented a FAP fraction marginally above the 0.08 threshold, but
visually displayed a trend dissimilar to other post-rain periods and so was re-categorized as Dry. Sample
28 showed no obvious rainfall, but the measurement team observed persistent fog in three consecutive
mornings (Samples 25, 27, 28), and the concentration of fluorescent (2-6 μm) particles suggested a source
of particles not influenced by rain, and so this Rainy sample was re-categorized as Other. Sample 38
displayed a fluorescent number ratio just below the threshold value, and was thus categorized as Dry,
however, the measurement team observed post-rain periods at the beginning and end of the sample, so the
sample were re-categorized as Other. For all samples other than these five, the categorization was
determined using the majority (> 0.50) of the 15 min periods. In no cases other than the five that were re-
categorized was the highest category fraction less than 0.50 of the sample time. Note that we have chosen
to capitalize Rainy, Dry, and Other to highlight that we have rigorously defined the period using the
characterization scheme described above and to separate the nomenclature from the general, colloquial
usage of the terms. Wetness category assignment for each high volume filter sample period is shown in
Figure 2 as a background color (brown for Dry samples, green for Rain–influenced samples, and pink for
Other samples) and Table S1.





To validate the qualitative differences between wetness categories described in the last section,
we present observations about each of these groupings. First, we organized the WIBS data according to
the particle categories introduced by Perring et al. (2015). By this method, every fluorescent particle
detected by the WIBS can be defined uniquely into one of seven categories (i.e. A, AB, ABC and so on).
By plotting the relative fraction of fluorescent particles described by each particle type, temporal
differences between measurement periods can be observed, as shown in Figure 2e. To a first
approximation, this analysis style allows for coarse discrimination of particle types. For example, a given
population of particles would ideally exhibit a consistent fraction of particles present in the different
particle categories as a function of time. By this reasoning, sample periods categorized as Dry (most of
the latter half of the study; brown bars in Fig. 2) would be expected to have a self-consistent particle type
trend, whereas sample periods categorized as Rainy (most of the first half of the study; green bars in Fig.
2) would have a self-consistent particle type trend, but different from the Dry samples. This is broadly
true. During Rainy periods as seen in Figure 3a, there is a relatively high fraction ($> 65\%$) of ABC
particles (light blue) and a relatively low fraction ($< 15\%$) in BC (purple) and C (yellow) type particles,
suggesting heavy influence from the FL1 channel. In contrast, during Dry periods the fraction of ABC
particles (light blue) is reduced ($<25\%$) while BC (purple) and C (yellow) type particles increase in
relative fraction ($>30\%$ and $>40\%$, respectively) suggested a diminished influence of FL1 channel.
It is important to note a few important caveats here. First, the ability of the WIBS to discriminate
finely between PBAP types is relatively poor and it is still unclear exactly how different particle types
would appear by this analysis method. Particles of different kinds and from different sources are likely
convolved into a single WIBS particle type, which could either soften or enhance the relationships with
rain discussed here. Second, the assignment of particle types is heavily size-dependent and sensitive to
subtle instrument parameters, and so it is unclear how different instruments would present similar particle
types. For example, Hernandez et al. (2016) used two WIBS instruments and found differences in relative
fraction of particle categories for samples aerosolized in the lab. They reported fungal spores to be
predominately A, AB, and ABC type particles, whereas Rainy sample periods suggested to have heavy
fungal spore influence by Huffman et al. (2013) show predominantly C, BC, and ABC particle fraction.
These discrepancies may be due to the comparison of ambient particles to laboratory-grown cultures. The
highly controlled environment of a laboratory may not always accurately represent the humidity
conditions in which fungal spore release occurs in this forest setting (Saari et al., 2015). This would
impact the fluorescence properties of the fungal spore particles which are inhibited by increased moisture
level around the spore (Hill et al., 2009). More likely, however, is that the WIBS-3 used here exhibits
higher sensitivity in the FL3 channel with respect to the FL1 and FL2 channels (Robinson et al., 2013), as
compared to the WIBS-4A used one of the units reported by Hernandez et al. (2016). This would explain
the shift here towards particles with C-type fluorescence. One piece of evidence for this is the quantitative
comparison of particle measurements presented by the UV-APS and WIBS-3 instruments co-deployed
here (Fig. 4). The number concentration of particle exhibiting fluorescence above the FL2 baseline of the
WIBS-3 is approximately consistent with the number of fluorescent particles measured by the UV-APS,
and significantly below the concentration of FL3 particles. The UV-APS number concentration shows the
highest correlation with the WIBS FL2 channel: during Rainy periods, $R^2 = 0.70$; Dry, $R^2 = 0.82$; Other, $R^2$
$= 0.92$. These observations are in stark contrast to the trends reported by Healy et al. (2014) that the UV-
APS fluorescent particle concentration correlated most strongly with the WIBS-4 FL3 and that the
number concentration of FL3 was the lowest out of all three channels. Given that the FL3 channel of the
WIBS and the UV-APS probe cover similar excitation and emission wavelengths it is expected that these
two channels should correlate well. Based on these data, we suggest that the WIBS-3 utilized here may
present a very different particle type break-down than if a WIBS-4 had been used. So, while caution is
recommended when comparing the relative break-down of WIBS particle categories shown here (Fig. 3)
with other studies, the data are internally self-consistent, and comparing qualitative differences between,
e.g. Rainy and Dry periods is expected to be robust. The main point to be highlighted here is that there is





indeed a qualitative difference in particles present in the three wetness categories, as averaged and shown
in Figure 3a, which generally supports the effort to segregate these samples.

Further evidence that there is a qualitative difference in the three wetness categories is shown
using molecular genetic analysis (Figs. 3b, c). The analysis of fungal DNA sequences from 21 of the high
volume samples found 406 operational taxonomic units (OTUs), belonging to different fungal classes and
phyla. When organized by wetness type it was observed that 106 of these occurred only on Rainy
samples, 148 of these occurred on Dry samples, and 37 on Other samples, with some fraction occurring in
overlaps of each (Fig. 3c). This shows that the number of OTUs observed uniquely in either the Rainy or
Dry periods is greater than the number of OTUs present in both wetness types, suggesting that the fungal
communities in each grouping are relatively distinct. Further, Figure 3b shows a break-down of fungal
taxonomic groupings for each wetness group. This analysis shows that there is a qualitative difference in
taxonomic break-down between periods of Rainy and Dry. Specifically, during Dry periods there is an
increased fraction of Pucciniomycetes (green bar, Fig. 3c), Chytridiomycota (yellow), Sordariomyctes
(orange), and Eurotiomycetes (pink) when compared to the Rainy periods.

**3.2 Atmospheric mass concentration of arabitol, mannitol, and fungal spores**

To estimate fungal spore emission to the atmosphere, the concentration of arabitol and mannitol
(Fig. 5a, b, Table S2) in each aerosol sample was averaged for all samples in each of the three wetness
categories. The average TSP concentration of arabitol collected on Dry samples increased by a factor of
3.3 on Rainy samples ($35.2 \pm 10.5$ ng m$^{-3}$), and the average TSP mannitol concentration on Rainy samples
was higher by a factor of 3.7 ($44.9 \pm 13.8$ ng m$^{-3}$). Figures 5a, b show the concentration variability for
each wetness category, observed as the standard deviation from the distribution of individual samples. For
each polyol, there is no overlap in the ranges shown, including the outliers of the Rainy and Dry category,
suggesting a definitive and conceptually distinct separation between dry periods and those influenced by
rain. The concentrations observed during Other periods is between those of the Dry and Rainy averages,
as expected, given the difficulty in confidently assigning these uniquely to one of these categories. The
observations here are roughly consistent with previous reports of polyol concentration, despite differences
in local fungal communities and concentrations. For example, Rathnayake et al. (2016) observed 30.2 ng
m$^{-3}$ arabitol and 41.3 ng m$^{-3}$ mannitol in PM$_{10}$ samples collected in rural Iowa, USA. In addition, Zhang et
al. (2015) reported arabitol and mannitol concentrations in PM$_{10}$ samples of 44.0 and 71.0 ng m$^{-3}$,
respectively, from a study in the mountains on Hainan Island off the coast of Southern China.

The square of the correlation coefficient ($R^2$) here between concentration values of arabitol and
mannitol during Rainy samples is very high (0.839; Table 1) suggesting that arabitol and mannitol
originated from a primarily from the same source, likely active-discharge fungal spores.  The correlation
is similar to the 0.87 $R^2$ reported by Bauer et al. (2008a) and the 0.93 $R^2$ reported by Graham et al. (2003).
In contrast, the same correlation between mannitol and arabitol concentrations, but for Dry samples is
relatively low (0.312). This is consistent with reports that arabitol can be used more specifically as a spore
tracer, but that mannitol has additional atmospheric sources besides fungal spores. The same correlation
was also performed between arabitol or mannitol and other molecular tracers (endotoxins and $(1{\rightarrow}3)$-$\beta$-
D-glucan), but all $R^2$ value were less than 0.43, suggesting that the endotoxins and glucans analyzed were
not emitted uniquely from the same sources as arabitol and mannitol.

Results from the two UV-LIF instruments were averaged over high volume sample periods, and a
correlation analysis was performed between tracer mass and fluorescent particle mass showing positive
correlations in all cases. The FAP mass from the UV-APS shows high correlation with the fungal polyols
during Rainy periods, with $R^2$ of 0.732 and 0.877 for arabitol and mannitol, respectively (Table 2; Figure
5c, d). The same tracers correlate poorly with the UV-LIF during Dry conditions. This is expected,
because polyols such as arabitol and mannitol are only found in *Ascomycota* and *Basidiomycota* fungal
spores which both utilize wet discharge methods for spore dispersal (Elbert et al., 2007; Feofilova, 2001;





Lewis and Smith, 1967). This high correlation suggests that the UV-APS does a good job of detecting
these wet-discharge spores, and corroborates previous statements that particles detected are often
predominately fungal spores (Healy et al., 2014; Huffman et al., 2013; Huffman et al., 2012). In contrast,
the low slope value and the poor correlation during Dry periods suggest that the UV-APS is also sensitive
to other kinds of particles, as designed. The small positive x-offset (FAP mass; Table S2, Figs. 5c,d)
during Rainy periods is likely due to particles that are too weakly fluorescent to be detected and counted
by the UV-APS, which is consistent with observations made in Brazil (Huffman et al., 2012).
Particle mass from WIBS Cl1, assigned to fungal spores (Crawford et al., 2015), also correlated
strongly with the same two molecular tracers. Both Rainy periods ($R^2$ 0.824) and Dry periods ($R^2$ 0.764)
correlate well with arabitol (Fig. 5e), while mannitol (Fig. 5f) only shows a strong correlation during the
Rainy periods ($R^2$ 0.799). Mannitol is a common polyol in higher plants while arabitol is only found in
fungal spores and lichen (Lewis and Smith, 1967). So the strong correlation of each polyol with UV-LIF
mass during Rainy periods when actively-discharged spores are expected to dominate and the similarly
strong correlations associated with arabitol suggests that the Cl1 cluster does a reasonably good job of
selecting fungal spore particles. The poor correlation between mannitol and Cl1 during dry periods
illustrates that the background mannitol concentration is likely not due to fungal spores alone, but has
contribution from other higher plants that contain mannitol. Particle concentrations detected by individual
WIBS channels and in the other cluster were also compared with polyol concentrations, but each
correlation is relatively poor compared to that with respect to Cl1. As seen in Table 2 and Figures S2-S3,
correlations in FL1, 2, and 3 with arabitol are poor (<0.4) in the Dry category and good (0.4 < $R^2$ <0.7) in
the Rainy category. For mannitol, all the UV-LIF instruments show high correlation (>0.7) in all cases.
This is likely due to mannitol being a non-specific tracer and suggests that the majority of UV-LIF
particles observed during all periods was dominated by PBAP.
### 3.3 Estimated number concentration of fungal spore aerosol
Bauer et al. (2008a) reported measurements of fungal spore number concentration in Vienna,
Austria using epifluorescence microscopy and also measured fungal tracer mass concentrations in order to
estimate the mass of arabitol (1.2 to 2.4 pg spore$^{-1}$) and mannitol (0.8 to 1.8 pg spore$^{-1}$) associated with
each emitted spore. Bauer et al. (2008a) and (Yttri et al., 2011b) reported ratios of mannitol to arabitol of
approximately 1.5 (± standard deviation of 26%) and 1.4 ± 0.3, respectively. Our measurements show
slightly lower ratios of mannitol to arabitol, but that the ratio is dependent on wetness category; Rainy,
1.29 ± 0.17; Dry, 1.12 ± 0.23; and Other, 1.24 ± 0.54. The mannitol to arabitol ratio would be expected to
vary as a function of fungal population present in the aerosol, whether between different wetness periods
at a given location or between different physical localities.
Using the approximate mid-point of the Bauer et al. (2008a) reported ranges, 1.7 pg mannitol per
spore and 1.2 pg arabitol per spore, atmospheric number concentrations of spores collected onto the high
volume filters were calculated from the polyol mass concentrations measured here. Based on these values,
and assuming all polyol mass originated with spore release, the mass concentration averages (Fig. 5) were
converted to fungal spore number concentrations (Fig. 6). The trends of spore concentration averages are
the same as with the polyol mass, because the numbers were each multiplied by the same scalar value.
After doing so, the analysis reveals an estimated spore concentration during Dry periods of 0.89 x $10^4$ (±
0.21) spores m$^{-3}$ using the arabitol concentration and 0.70 x $10^4$ (± 0.19) spores m$^{-3}$ using the mannitol
concentration (Table 3). The estimated concentration of spores increased approximately three-fold during
Rainy periods to 2.9 x $10^4$ (± 0.8) spores m$^{-3}$ (arabitol estimate) and 2.6 x $10^4$ (± 0.8) spores m$^{-3}$ (mannitol
estimate) (Figure 6a, b). These estimates match well with estimates reported by Spracklen and Heald
(2014) , who modeled the concentration of airborne fungal spores across the globe as an average of 2.5 x
$10^4$ spores m$^{-3}$, with approximately 0.5 x $10^4$ spores m$^{-3}$ over Colorado.



The UV-LIF instruments discussed here are fundamentally number-counting techniques and can be utilized here roughly as spore counters. As a first approximation, each particle detected by the UV-APS was assumed to be a fungal spore with the same properties used in the assumptions by Bauer et al. (2008a). Plotting the correlation of fungal spore number concentration from polyol mass concentration with respect to the fungal spore concentration assumed from the UV-LIF measurements shows correlations in Figures 6c-f. The first, and most important observation is that the estimated fungal spore concentration from each technique is on the same order of magnitude, $10^4$ m$^{-3}$. Looking at individual correlations reveals a finer layer of details. These results show that the number concentration of fungal spores estimated by the UV-APS is greater than the number of fungal spores estimated by the tracers, as evidenced by slope values of approximately 0.2 and 0.35 for Rainy and Dry conditions, respectively (Figure 6c, d). The $R^2$ values (~0.5) during Rainy periods indicate that the additional source of particles detected by the UV-APS is likely to have a similar source, such as PBAP mechanically ejected from soil and vegetative surfaces with rain-splash (Huffman et al., 2013). The magnitude of the over-estimation is higher during Dry periods, which would be expected if Rainy periods exhibited much higher particle number fractions associated with polyol-containing spores.

The Cl1 cluster from WIBS data shows correlations with estimated fungal spores from arabitol and mannitol that have slope much closer to 1.0 than correlations with UV-APS number (Figure 6e, f, Table S3). For example, the slope of the Cl1 correlations with each polyol during Rainy periods is approximately 0.87. This suggests only a 13% difference between the spore concentration estimates from the two techniques during Rainy periods. The average number concentration of Cl1 during Rainy periods is 1.6 x $10^4$ (± 0.8) spores m$^{-3}$. In both cases the slopes with respect to Cl1 is greater than 1.0 during Dry periods, suggesting that the cluster method may be missing some fraction of weakly fluorescent particles. Huffman et al. (2012) similarly suggests that that particles that are weakly fluorescent may be below the detection limit of the instrument, and Healy et al. (2014) suggested that both UV-APS and WIBS-4 instruments significantly under-count the ubiquitous *Cladosporium* spores that are most common during dry weather and often peak in the afternoon when RH is low (De Groot, 1968; Oliveira et al., 2009). Fundamentally, however, the results from the UV-APS, and even more so the numbers reported by the clustering analysis by Crawford et al. (2015), reveal broadly similar trends with the numbers estimated from polyol-to-spore values reported by Bauer et al. (2008a).

The fungal culture samples show similar division during Rainy and Dry periods as arabitol and mannitol concentrations (Figure 6c), with an increase of approx. 1.6 during Rainy periods. The trend of a positive slope with respect to the UV-LIF measurements is also similar between the tracer and culturing methods. In general, however, the $R^2$ value correlating CFU to fungal spore number calculated from UV-LIF number is lower than between tracers and UV-LIF numbers (Tables 2, S4). This is not unexpected for several reasons. First, the short sampling time of the culture samples (20 min) leads to poor counting statistics and high number concentration variability, whereas each data point from the high volume air samples represents a period of 4 – 48 hours. Second, culture samplers, by their nature, only account for cultural fungal spores. It has been estimated that as low as 17% of aerosolized fungal spores are culturable, and so it is expected that the CFU concentration observed is significantly less than the total airborne concentration of spores (Bridge and Spooner, 2001; Després et al., 2012). Nonetheless, the culturing analysis here supports the tracer and UV-LIF analyses and the most important trends are consistent between all analysis methods. The concentration of fungal spores is higher during the Rainy periods, and there is a positive correlation between both tracer and CFU concentration and UV-LIF number.

In pristine environment, such as the Amazon, supermicron particle mass has been found to consist of up to 85% biological material (Pöschl et al., 2010). Total particulate matter mass was calculated here from the UV-APS number concentrations (m$^{-3}$) and converted to mass for particles of aerodynamic diameter 0.5 – 15 μm. In only this case a density of 1.5 g cm$^{-3}$ was utilized to calculate a first

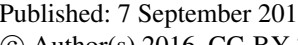



approximation of total particle mass to which all other mass measurements were compared. An average
TSP mass density of 1.5 g cm$^{-3}$ was utilized, because organic aerosol is typically estimated with density <
1.0 g cm$^{-3}$, biological particles are often assumed to have ca. 1.0 g cm$^{-3}$ density, and mineral dust
particles have densities of up to ca. 3.5 g cm$^{-3}$ (Dexter, 2004; Tegen and Fung, 1994). Fungal spore mass
was estimated here using the fungal spore concentrations calculated from arabitol and mannitol mass (Fig.
6) and then using an estimated 33 pg reported by Bauer et al. (2008b) as an average mass per spore.
Dividing the resultant fungal spore mass by total particulate mass provides a relative mass fraction for
each high volume sample period. These calculations suggest that fungal spores represent approximately
23% $\pm$ 9 (using arabitol) or 21% $\pm$ 8 (using mannitol) of total particulate mass during Rainy periods
(Table 3, Figure 7). This represents a nearly 6 fold increase in percentage compared to Dry periods (4.8%
$\pm$ 1.4 and 3.7% $\pm$ 1.1, respectively). A similar increase during Rainy periods was also seen in the mass
fraction of fungal cluster Cl1, which represented 17% $\pm$ 10 of the particle mass during Rainy and 2% $\pm$ 1
during Dry periods (Table S4).

**3.5 Variations in endotoxin and glucan concentrations**

Endotoxins are components of gram-negative bacteria (Andreae and Crutzen, 1997). Here, we
show correlations between total endotoxin mass and WIBS Cl$_{Bact}$, which were assigned by Crawford et al.
(2015) to be bacteria due to the small particle size (< 1 µm) and high correlation with rain. These cluster
assignments are quite uncertain, however, and should be treated loosely. The correlation between
endotoxin mass and UV-APS and the WIBS clusters was very poor, in most cases $R^2$ <0.1 (Table 2,
Figure 8), suggesting no apparent relationship. Analysis of bacteria by both UV-LIF techniques is
hampered by the fact that bacteria can be < 1 µm in size and because both instruments detect particles
with decreased efficiency at sizes below 0.8 µm. So weak correlations may not have been apparent due to
reduced overlap in particle size. Despite the lack of apparent correlation between the techniques, the
relatively variable endotoxin concentrations were elevated during Rainy periods, consistent with Jones
and Harrison (2004), who showed that bacteria concentration were elevated after rainy periods.

Glucans, such as (1→3)-β-D-glucan, are components of the cell walls of pollen, fungal spores,
plant detritus, and bacteria (Chow et al., 2015b; Lee et al., 2006; Stone and Clarke, 1992). In contrast to
the observed difference in endotoxin concentration during the different wetness periods, however, (1→3)-
β-D-glucan showed no correlations with UV-LIF concentrations (Table 2) and no differentiation during
the different wetness periods.

**4. Conclusions**

Increased concentrations of fluorescent aerosol particles and ice nuclei attributed to having
biological origin were observed during and immediately after rain events throughout the BEACHON-
RoMBAS study in 2011 (Huffman et al., 2013; Prenni et al., 2013; Schumacher et al., 2013). Here we
expand upon the previous reports by utilizing measurements from two commercially available UV-LIF
instruments, of several molecular tracers extracted from high volume filter samples, and from a culture-
based sampler in order to compare three very different methods of atmospheric fungal spore analysis.
This study represents the first reported correlation of UV-LIF and molecular tracer measurements and
provided an opportunity to understand how an important class of PBAP might be influenced by periods of
rainy and dry weather. We found clear patterns in the fungal molecular tracers, arabitol and mannitol,
associated with Rainy conditions that are consistent with previous findings (Bauer et al., 2008a; Elbert et
al., 2007; Feofilova, 2001). Fungal polyols increased 3-fold over Dry conditions during Rainy weather
samples, with arabitol concentration of 35.2 $\pm$ 10.5 ng m$^{-3}$ and mannitol concentration of 44.9 $\pm$ 13.8 ng
m$^{-3}$. Additionally, the very high correlation of the fungal tracers with WIBS Cl1 ($R^2 > 0.8$ in many cases)
provides support for its assignment by Crawford et al. (2015) to fungal spores. Similarly, the UV-APS
correlates well with fungal tracers, however over-counts the number concentration estimated from the
tracers, confirming that the UV-APS is sensitive also to other types of particles beyond fungal spores, as
expected. The estimated spore count from the WIBS Cl1 concentration was within ~13% of the spore
count estimated by the tracer method, with concentrations ranging from 1.6 – 2.9 x 10$^4$ spores m$^{-3}$. These



values are broadly consistent with concentrations modeled by, e.g. Spracklen and Heald (2014), Hoose et al. (2010), and Hummel et al. (2015). These spore counts represent 17-23% of the total particle mass during Rainy conditions and 2-4% during Dry conditions. Culture-based sampling also shows a similar relationship between CFU and UV-LIF concentrations and an increase of ~1.6 between Dry and Rainy conditions. Despite the fact that the tracer and UV-LIF approaches to estimating atmospheric fungal spore concentration are fundamentally different, they provide remarkably similar estimates and temporal trends. With further improvements in instrumentation and analysis methods (e.g. advanced clustering algorithms applied to UV-LIF data), the ability to reliably discriminate between PBAP types is improving. As we have shown here, this technology represents a potential for monitoring approximate fungal spore mass and for contributing improved information on fungal spore concentration to global and regional models that to this point has been lacking (Spracklen and Heald, 2014).

**5. Acknowledgements**

The BEACHON-RoMBAS campaign was partially supported by an ETBC (Emerging Topics in Biogeochemical Cycles) grant to the National Center for Atmospheric Research (NCAR), the University of Colorado, Colorado State University, and Penn State University (NSF ATM-0919189). The authors wish to thank Jose Jimenez, Douglas Day (Univ. Colorado-Boulder); Anthony Prenni, Paul DeMott, Sonia Kreidenweis, and Jessica Prenni (Colorado St. Univ.); Alex Guenther, and Jim Smith (NCAR) for BEACHON-RoMBAS project organization and logistical support and the USFS, NCAR, and Richard Oakes for access to the Manitou Experimental Forest Observatory field site. Measurements of temperature, relative humidity, wind speed, and wind direction were provided by Andrew Turnipseed (NCAR) and leaf wetness and disdrometer data were provided by Dave Gochis (NCAR). Marie I. Gosselin thanks the Max Planck Society for financial support. J. Alex Huffman thanks the University of Denver for intramural funding for faculty support. The Mainz team acknowledges financial support from the Max Planck Society (MPG), the Max Planck Graduate Center with the Johannes Gutenberg University Mainz (MPGC), the Geocycles Cluster Mainz (LEC Rheinland-Pfalz), and the German Research Foundation (DFG PO1013/5-1 and FR3641/1-2, FOR 1525 INUIT). The Manchester team acknowledges funding from the UK NERC (UK-BEACHON, Grant # NE/H019049/1) to participate in the BEACHON experiment, and development support of the WIBS instruments. Manchester would also like to thank Prof. Paul Kaye, the developer of the WIBS instruments and his team at the University of Hertfordshire, for their technical support. The authors thank Cristina Ruzene, Isabell Müller-Germann, Petya Yordanova, Tobias Könemann (Max Planck Inst. For Chem.), and Nicole Savage (Univ. Denver) for technical assistance.



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





**Tables and Figures:**

| | | | Mass Concentration | | | | | |
|---|---|---|---|---|---|---|---|---|
| | | | Arabitol (ng m$^{-3}$) | | Mannitol (ng m$^{-3}$) | | (1→3)-β-D-glucan (pg m$^{-3}$) | |
| | | | Rainy | Dry | Rainy | Dry | Rainy | Dry |
| Mass Concentration | Mannitol (ng m$^{-3}$) | Rainy | **_0.839_** | | | | | |
| | | Dry | | 0.312 | | | | |
| | (1→3)-β-D-glucan (pg m$^{-3}$) | Rainy | 0.000 | | 0.003 | | | |
| | | Dry | | 0.000 | | 0.327 | | |
| | Endotoxins (EU m$^{-3}$) | Rainy | 0.116 | | 0.126 | | **0.427** | |
| | | Dry | | 0.012 | | 0.113 | | 0.103 |

**Table 1**: Square of correlation coefficients ($R^2$) comparing total mass concentration of molecular tracers
to each other. EU: endotoxin units. Boxes colored by coefficient value (**_Bold Underline_**> 0.7; 0.7 > **Bold**
> 0.4).





| | | Mass Concentration | | | | | | | | Fungal Spore Number Conc. | | | | | |
| | | Arabitol (ng m³) | | Mannitol (ng m⁻³) | | (1→3)-β-D-glucan (pg m⁻³) | | Endotoxins (EU m⁻³) | | Arabitol (spores m⁻³) | | Mannitol (spores m⁻³) | | Colony Forming Units (CFU m⁻³) | |
| | | Rainy | Dry | Rainy | Dry | Rainy | Dry | Rainy | Dry | Rainy | Dry | Rainy | Dry | Rainy | Dry |
| UVAPS | | **_0.732_** | 0.127 | **_0.877_** | 0.160 | 0.006 | 0.012 | 0.153 | 0.067 | **0.483** | 0.278 | **0.504** | **0.571** | **0.469** | **0.491** |
| WIBS | FL | **0.554** | 0.250 | **_0.810_** | 0.255 | 0.128 | 0.010 | 0.068 | 0.066 | 0.159 | 0.200 | 0.088 | 0.314 | 0.330 | **_0.737_** |
| | FL1 | **0.602** | **0.445** | **_0.819_** | **0.412** | 0.042 | 0.001 | 0.090 | 0.012 | **0.667** | 0.339 | **_0.863_** | **0.621** | **0.470** | **0.546** |
| | FL2 | **0.617** | 0.248 | **_0.843_** | 0.342 | 0.092 | 0.001 | 0.039 | 0.094 | **0.485** | 0.302 | **0.442** | 0.340 | **0.560** | **0.543** |
| | FL3 | **0.561** | 0.222 | **_0.818_** | 0.251 | 0.124 | 0.008 | 0.071 | 0.065 | 0.178 | 0.181 | 0.104 | 0.306 | 0.367 | **_0.736_** |
| | Cl1 | **_0.824_** | **_0.764_** | **_0.799_** | 0.109 | 0.000 | 0.134 | 0.229 | 0.011 | **0.679** | **0.543** | **_0.775_** | **0.423** | 0.128 | **0.690** |
| | Cl2 | 0.005 | 0.002 | 0.004 | 0.006 | 0.002 | 0.047 | 0.006 | 0.017 | 0.052 | 0.056 | 0.001 | 0.075 | 0.081 | **_0.930_** |
| | Cl3 | 0.267 | 0.164 | 0.261 | 0.198 | 0.003 | 0.011 | 0.016 | 0.066 | 0.052 | 0.116 | 0.087 | **0.439** | 0.262 | 0.383 |
| | Cl4 | 0.048 | 0.046 | 0.172 | 0.118 | 0.115 | 0.011 | 0.179 | 0.145 | 0.062 | 0.089 | 0.001 | 0.065 | 0.120 | 0.000 |
| | Cl_Bact | | | | | | | 0.041 | 0.081 | | | | | | |

(UV-LIF Mass or Number Concentration)

**Table 2**: Square of correlation coefficients ($R^2$) comparing fluorescent particle measurements from UV-LIF instruments to measurements from molecular tracers. Columns marking tracer mass (top line) indicate correlations between time-averaged UV-LIF and tracer mass concentrations (left side), and columns marking fungal spore number indicate correlations between fungal spore number concentrations estimated from time-averaged UV-LIF and tracer or culture measurements (right side). FL1, FL2, FL3 represent individual channels from the WIBS. FL represents all particle exhibiting fluorescence in any channel. Cl1, Cl2, Cl3, Cl4 are clusters that estimate particle concentrations as a mixture of various channels (Crawford et al., 2015). Cl_Bact is a sum of the "bacteria" clusters Cl2-4. Boxes colored by coefficient value (**Bold Underline** > 0.7; 0.7 > **Bold** > 0.4).



| | Mass Concentration | | | | | | |
|---|---|---|---|---|---|---|---|
| | Arabitol (ng m⁻³) | Mannitol (ng m⁻³) | Erythritol (ng m⁻³) | Levoglucosan (ng m⁻³) | Glucose (ng m⁻³) | Endotoxins (EU m⁻³) | (1→3)-β-D-glucan (pg m⁻³) |
| Dry | 10.6 ± 2.5 n = 18 | 11.9 ± 3.2 n=18 | 0.840 ± 0.610 n=16 | 14.2 ± 10.7 n=15 | 38.7 ± 21.3 n=18 | 0.192 ± 0.0970 n=18 | 8.8 5 ± 7.68 n=18 |
| Rainy | 35.2 ± 10.5 n=11 | 44.9 ± 13.8 n=11 | 1.12 ± 0.38 n=3 | 12.4 ± 19.1 n=8 | 73.2 ± 50.5 n=11 | 1.43 ± 1.22 n=10 | 10.6 ± 8.2 n=11 |
| Other | 20.2 ± 8.9 n=6 | 22.7 ± 8.3 n=6 | 0.664 ± 0.515 n=6 | 9.21 ± 1.66 n=5 | 56.5 ± 39.2 n=6 | 0.311 ± 0.159 n=6 | 6.08 ± 6.08 n=6 |
| | Mass Contribution (%) | | | | | | |
| Dry | 0.18 % ± 0.05 n=18 | 0.202 % ± 0.073 n=18 | 0.0.14 % ± 0.011 n=16 | 0.21 % ±0.17 n=15 | 0.67 % ±0.49 n=18 | | 0.16 % ±0.16 n=18 |
| Rainy | 0.83 % ± 0.32 n=11 | 1.07 % ±0.44 n=11 | 0.032 % ±0.009 n=3 | 0.27 % ±0.41 n=8 | 1.60 % ±1.09 n=11 | | 0.25 % ±0.21 n=11 |
| Other | 0.25 % ± 0.28 n=6 | 0.37 % ± 0.29 n=6 | 0.013 % ±0.015 n=6 | 0.15 % ±0.11 n=5 | 0.83 % ±0.64 n=6 | | 0.12 % ±0.19 n=6 |
| | Fungal Spore Number Concentration (m⁻³) | | | | | | |
| Dry | 8870 ± 2060 n=18 | 6890 ± 1870 n=18 | | | | | |
| Rainy | 29310 ± 8727 n=11 | 26430 ± 8139 n=11 | | | | | |
| Other | 16850 ± 7415 n=6 | 13350 ± 4863 n=6 | | | | | |
| | Fungal Spore Mass Contribution (%) | | | | | | |
| Dry | 4.81 % ± 1.36 n=18 | 3.72 % ± 1.12 n=18 | | | | | |
| Rainy | 22.88 % ±8.84 n=11 | 20.66 % ±8.49 n=11 | | | | | |
| Other | 9.80 % ± 7.67 n=6 | 7.31 % ± 5.60 n=6 | | | | | |

**Table 3:** Campaign-average concentrations of molecular tracers (measured) and fungal spores (number
concentration estimated from arabitol and mannitol mass). Each set of data broken into wetness
categories. Values are mean ± standard deviation; *n* shows the number of samples used for averaging.
Fungal spore mass contribution was based on the assumption by Bauer et al. (2008b) of 33 pg spore⁻¹.
Total particulate matter mass calculated from UV-APS number concentration (m⁻³) and converted to mass
over aerodynamic particle diameter range 0.5 – 15 μm using density of 1.5 g cm⁻³.

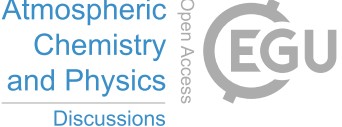




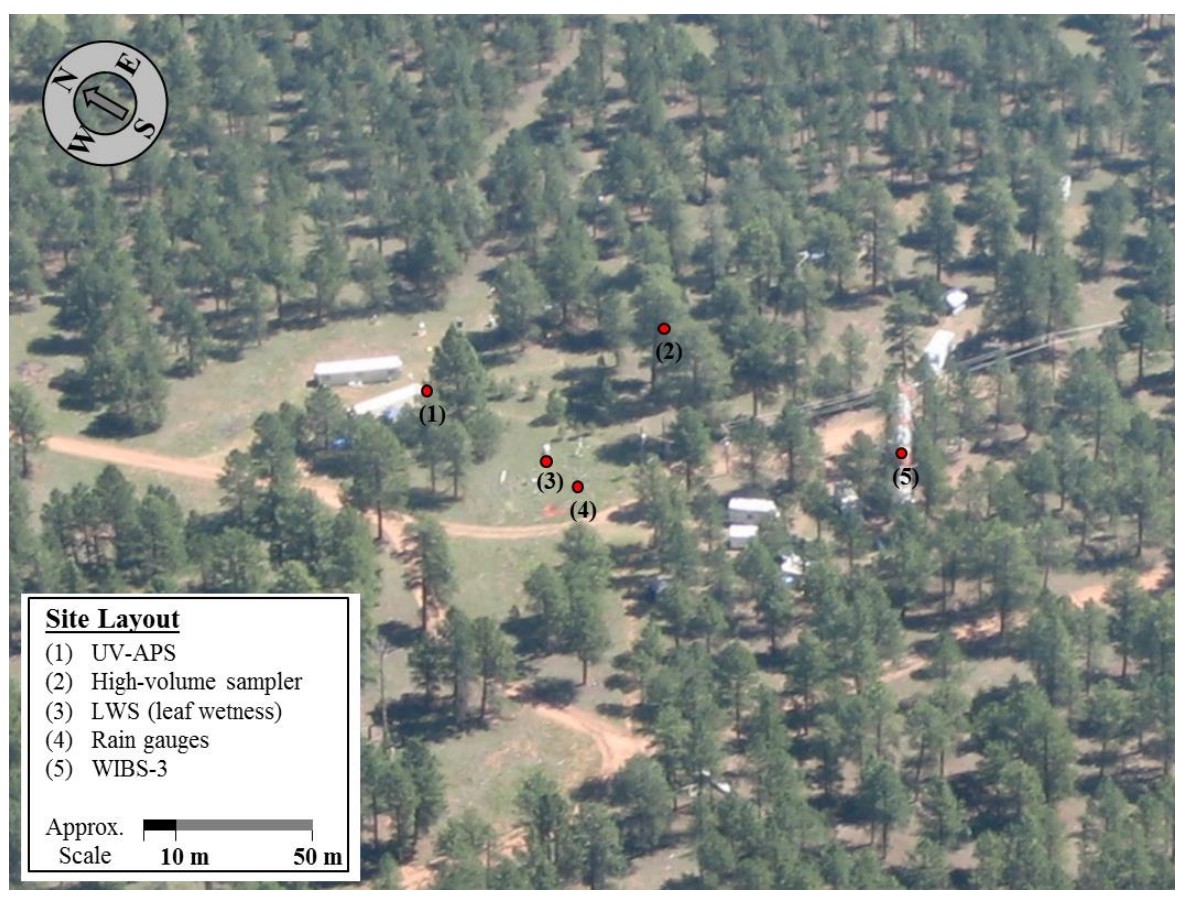

**Figure 1:** Aerial overview of BEACHON-RoMBAS field site at the Manitou Experimental Forest
Observatory located northwest of Colorado Springs, CO. Locations of all instruments and sensors
discussed here are marked and were located within a 50 m radius. Figure adapted from Figure 1a of
Huffman et al. (2013)

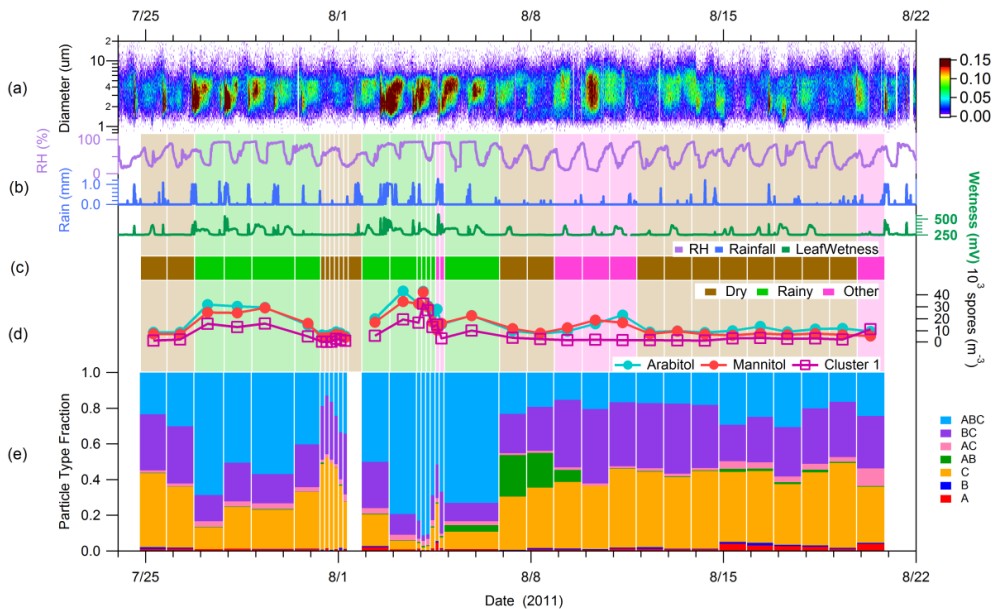

**Figure 2**: Time series of key species concentrations and meteorological data over entire campaign. (a)
Fluorescent particle number size distribution measured with UV-APS instrument. Color scale indicates
fluorescent particle number concentration (L$^{-1}$). (b) Meteorological data: relative humidity (RH),
disdrometer rainfall (mm per 15 min), leaf wetness (mV). (c) Wetness category indicated as colored bars;
green, Rainy; brown, Dry; pink, Other. Bar width corresponds to filter sampling periods. Lightened
colored bars extend vertically to highlight categorization. (d) Colored traces show fungal spore
concentrations estimated from molecular tracers (circles) and WIBS Cl1 data (squares). I Stacked bars
show relative fraction of fluorescent particle type corresponding to each WIBS category.





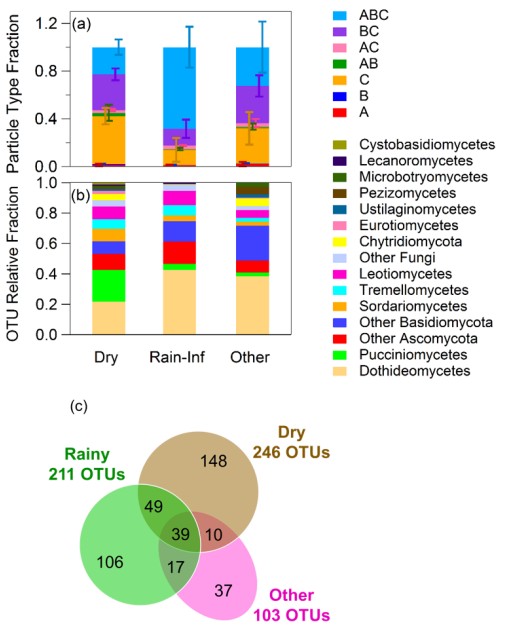

**Figure 3:** Characteristic differences between different wetness periods (Dry, Rainy, Other). (a) Relative
fraction of fluorescent particle number corresponding to each WIBS category. Bars show relative standard
deviation of category fraction in each wetness group (Dry, 19 samples; Rainy, 11 samples; Other, 6
samples). (b, c) Distribution of fungal OTU (operational taxonomic unit) values. (b) Fungal community
composition at phylum and class level with Agaricomycetes (dominant class with consistently ~60% of
diversity) removed. Relative proportion of OTUs assigned to different fungal classes and phyla for each
sample category shown. (c) Venn diagram showing the number of unique (wetness category specific) and
shared OTUs (represented by numbers in overlapping areas) among the sample categories (Dry, 11
samples; Rainy, 7 samples; Other, 3 samples). OTUs classified as cluster of sequences with ≥ 97%
similarity. Taxonomic assignments were performed using BLAST against NCBI database. In total, 3902
sequences, representing 406 fungal OTUs from 3 phyla and 12 classes were detected. Despite differences
in community structure across the sample categories, phylogenetic representation appears largely similar.





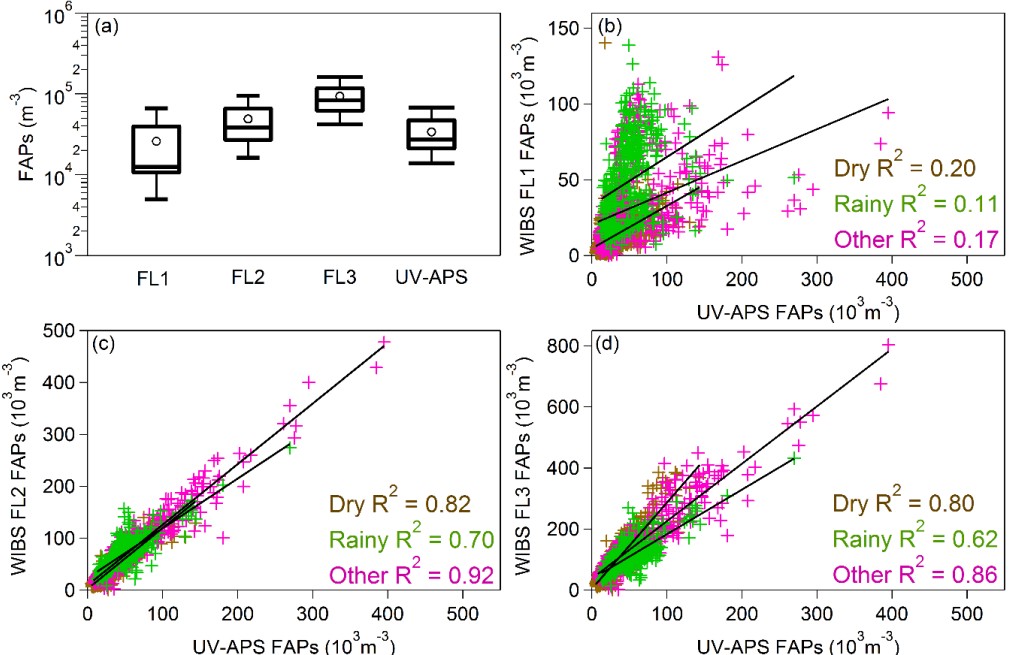

**Figure 4**: Number concentration of fluorescent particles as a function of instrument channel, averaged
over entire measurement period. (a) Box-whisker plot of fluorescent particle number concentration for
WIBS FL1, FL2, FL3, and UVAPS. Circle markers shows mean values, internal horizontal line shows
median, top and bottom of box show inner quartile, and whiskers show 5[th] and 95[th] percentiles. (b) WIBS
FL1 versus UV-APS (c) WIBS FL2 versus UV-APS (d) WIBS FL3 versus UV-APS. Crosses represent 5-
minute average points. Linear fits assigned for data in each wetness category.





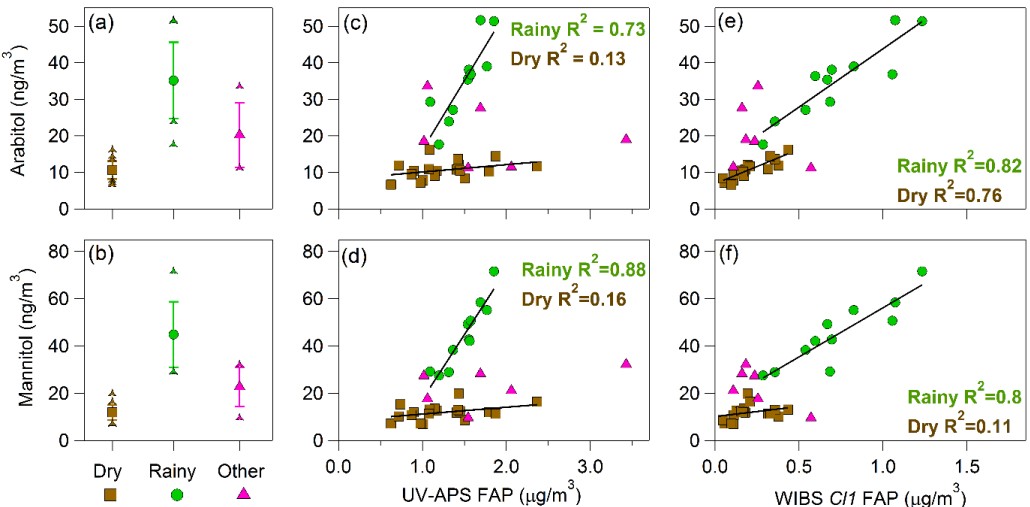

**Figure 5**: Mass concentrations of molecular tracers and fluorescent particles (assuming unit density
particle mass): arabitol – top row, and mannitol – bottom row. Average mass concentration of arabitol (a)
and mannitol (b) in each wetness category. Central marker shows mean value of individual filter
concentration values, bars represent standard deviation ($s$) range of filter values, and individual points
show outliers beyond mean $\pm\ s$. Correlation of arabitol (c) and mannitol (d) with fluorescent particle mass
from UV-APS. Correlation of arabitol I and mannitol (f) with fluorescent particle mass from WIBS
Cluster 1. $R^2$ values shown for each fit in c, d, e, f. Linear fit parameters are shown in Table S2.

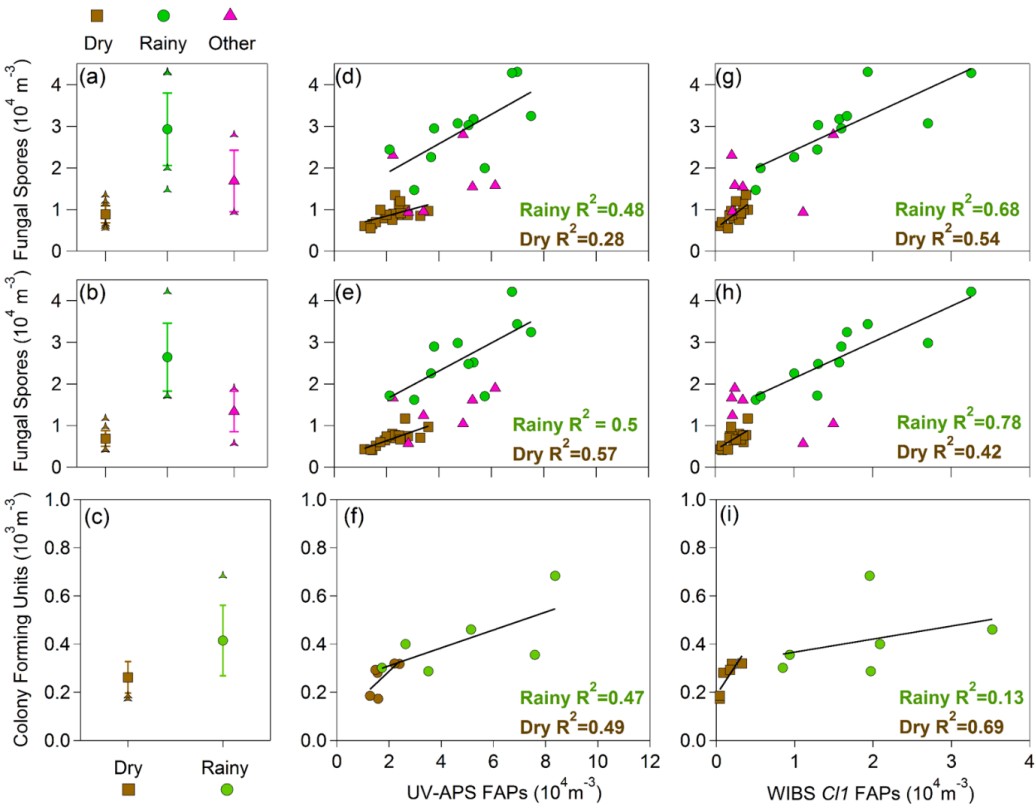

**Figure 6**: Estimated fungal spore number concentration, calculated using mass of arabitol and mannitol per spore reported by Bauer et al. (2008a). Estimates from arabitol (top row) and mannitol (middle row). Average fungal spore concentration, calculated using arabitol mass (a), mannitol mass (b), and colony forming units (c) in each wetness category. Central marker shows mean value of individual filter concentration values, bars represent standard deviation ($s$) range of filter values, and individual points show outliers beyond mean $\pm$ $s$. Correlation of fungal spore number calculated from arabitol (d), mannitol (e), and colony forming units (f) concentration with estimated fluorescent particle mass from UV-APS. Correlation of fungal spore number calculated from arabitol (g), mannitol (h), and colony forming unit (i) concentration with fluorescent particle concentration from WIBS Cluster 1. $R^2$ value shown for each fit (right two columns). Linear fit parameters are shown in Table S3.




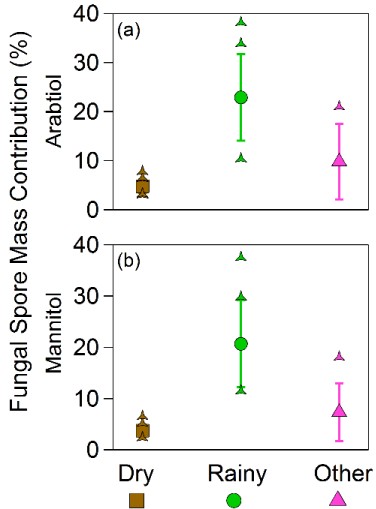

**Figure 7:** Estimated fraction of total aerosol mass contributed by fungal spores. Fungal spore mass
concentration ($\mu$g/m$^3$) calculated separately from mannitol and arabitol concentration and using average
mass per spore reported by Bauer et al. (2008b). Total particulate matter mass calculated from UV-APS
number concentration (m$^{-3}$) and converted to mass over aerodynamic particle diameter range $0.5 - 15$ $\mu$m
using density of 1.5 g cm$^{-3}$. Central marker shows mean value of individual filter concentration values,
bars represent standard deviation ($s$) range of filter values, and individual points show outliers beyond
mean $\pm$ $s$.




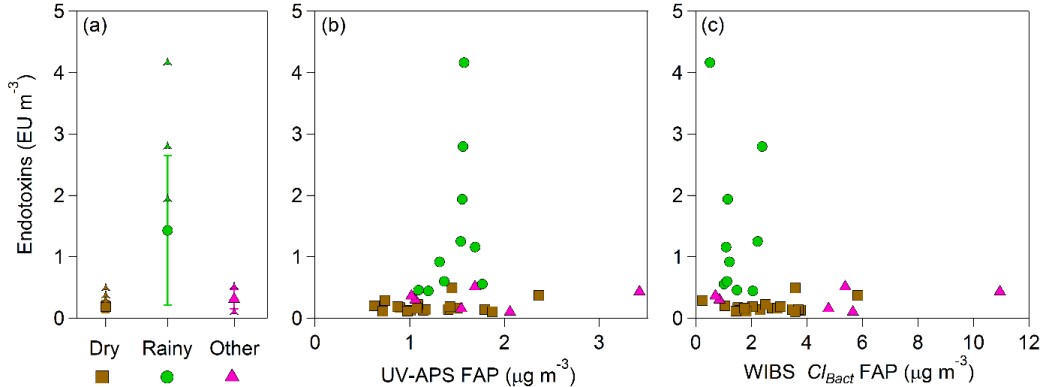

**Figure 8**: Endotoxin mass concentration as an approximate indicator of gram-negative bacteria
concentration. (a) Averaged concentration in each wetness category. Central marker shows mean value of
individual filter concentration values, bars represent standard deviation (*s*) range of filter values, and
individual points show outliers beyond mean ± *s*. (b) Correlation of endotoxin mass concentration with
estimated fluorescent particle mass from UV-APS. (c) Correlation of endotoxin mass concentration with
estimated fluorescent particle mass summed from Clusters 2, 3, and 4 from Crawford et al. (2015).