# Peer review of "Title: Fluorescent Bioaerosol Particle, Molecular Tracer, and Fungal Spore Concentrations during Dry"

_Atmospheric Chemistry and Physics, 2016_

## Referee Comment (RC1) · Anonymous Referee #3 · 5 Oct 2016

The manuscript is very well written and I believe of great relevance to the bioaerosol scientific community. The authors present very interesting and novel work comparing data from modern Light/Laser induced fluorescence (LIF) instruments with molecular tracers such as arabitol and mannitol. The paper also attempts to display the data in new ways scaling particle number to mass concentrations. The paper is very well cited and builds well on previous work. Thus I believe the paper should be published upon the correction of some minor technical/specific issues discussed below.

Specific/technical comments:

Comment 1: L62-63 "For example, asthma and allergies have shown notable increases during thunderstorms due to elevated bioaerosol concentrations" This is indeed true

however allergic rates have been climbing in recent years and I feel this should be incorporated. I suggest using the reference.

Linneberg, A., 2011. The increase in allergy and extended challenges. Allergy, 66(s95), pp.1-3.

Comment 2: L139 should $H_2O$ have a sub-scripted 2

Comment 3: Were the differences in sampling lines of the WIBS and UV-APS calculated? Reynolds number for instance?

Comment 4: Were all particles assumed to be spherical for the density calculations or was the WIBS ability to determine shape utilized?

Comment 5: Do you believe that cluster 1 is solely a fungal spore cluster, given its size range overlaps with that of some bacteria?

Comment 6: Why was a rainfall accumulation threshold of greater than 0.201 chosen?

Comment 7: What did the correlations look like before the manual reclassification of some of the rain/dry periods? How much did this effect it?

Comment 8: L 430-431. The Hill 2009 reference does talk about increased wetness effecting the fluorescent properties in comparison to dry samples however in this study wet samples were particles suspended in solution rather than particles at higher relative humidity's. I believe that this line should be rewritten. Do you believe the particles sampled during wet periods to be in droplets or to have increased moisture content? Could a moistened PBAP have increased fluorescence due to fluorescent compounds being extracted/leached to its surface?

Comment 9: Was there much difference in fluorescent intensity for FAP on Dry and Wet periods?

Comment 10:L555 Should "Figures 6 c-f" read "Figures 6 d-f"?

L560 Should "Figure 6 c, d" read "Figure 6 d, e"?

L567 Should "Figure 6 e, f" read "Figure 6 g, h"?

Comment 11: For the total particulate matter mass concentrations why did you not use the high volume sampler samples to determine the total mass? Instead of the UV-APS measurements.

Comment 12: You mention Cladosporium are generally present/released at dry periods was there any evidence that this occurred during this campaign?

---

## Referee Comment (RC2) · Anonymous Referee #4 · 6 Oct 2016

The article is of high quality providing a novel information relevant to the ACP addressing the atmospheric biological (fungal) tracers. The novelty is in correlations reported for periods affected by rain between fungal biomarkers obtained from offline measurements and fluorescent aerosol particle concentrations obtained by direct online measurements. The description of experimental work is sound and detailed supporting the good quality of the paper.

In my opinion, the article would gain if additional data with regard to total PM mass concentrations were reported. For example Table 3 presents % contribution of biomarkers with regard to particulate matter and spore mass. The estimated PM mass data presented along with the rest of the data would help to clarify relationship to overall chem-

ical characterization of PM if The data reported are comprehensive. Still are there also data available for the same period reporting on the occurrence of organic carbon and thus allowing for discussion of traditionally reported chemical characterization of organic particulate matter? Authors report on taxonomic differences in fungal DNA during wet and dry periods. Could such differences be attributed to the ability of different fungal species to survive in different humidity conditions?
* * *

---

## Referee Comment (RC3) · Anonymous Referee #2 · 13 Oct 2016

General Comments: The manuscript entitled "Fluorescent Bioaerosol Particle, Molecular Tracer, and Fungal Spore Concentrations during Dry and Rainy Periods in a Semi-Arid Forest" by Gosselin et al. reports correlations of fluorescent aerosol particles of UV-APS and WIBS-3 with molecular tracers of fungal spores and bacteria. This study provides further investigations of the detection ability of UV-LIF instruments of fungal spores. In general, the manuscript was well written and the analysis of the data was well performed. I recommend this manuscript to be accepted for publication after minor revisions.

Specific Comments: 1. In the last paragraph of Introduction and the Discussion sections, the authors declared that this is the first comparison of online UV-LIF with organic

molecular tracers measurements. In fact, a recent study has also made such comparisons between WIBS and fungal spore tracers (see Yue et al., 2016, Sci. Rep.). 2. In part 2.2 Online fluorescent instruments (Line 174 – 176), the fluorescent detection bands for WIBS-3 should be $\lambda$em 310 – 400 nm and $\lambda$em 400 – 600 nm (see Gabey et al., 2010, ACP). Please clarify it. 3. Line 205: Provide references for "One important difference between the models is that the WIBS-3 exhibits comparatively weak FL1 and FL2 signals with respect to the more updated models, and is thus more influenced by FL3". 4. In Figure 5 (e, f), the unit for WIBS Cl1 FAP was given as mass concentration. How do the authors convert the number concentrations to mass concentrations for WIBS-3? Such information should be provided in the Methods section.

---

## Author Comment (AC1) · 4 Nov 2016

> Note regarding document formatting: black text shows original referee comment, blue text shows author response, and red text shows quoted manuscript text. Changes to manuscript text are shown as *italicized and underlined*. All line numbers refer to discussion/review manuscript.

General Comments: The manuscript is very well written and I believe of great relevance to the bioaerosol scientific community. The authors present very interesting and novel work comparing data from modern Light/Laser induced fluorescence (LIF) instruments with molecular tracers such as arabitol and mannitol. The paper also attempts to display the data in new ways scaling particle number to mass concentrations. The paper is very well cited and builds well on previous work. Thus I believe the paper should be published upon the correction of some minor technical/specific issues discussed below.

> Author response: We thank the referee for his/her positive assessment and summary.

Specific Comments:
Comment 1: L62-63 "For example, asthma and allergies have shown notable increases during thunderstorms due to elevated bioaerosol concentrations" This is indeed true however allergic rates have been climbing in recent years and I feel this should be incorporated. I suggest using the reference. Linneberg, A., 2011. The increase in allergy and extended challenges. Allergy, 66(s95), pp.1-3.

> The Linneberg reference was added to L62.

Comment 2: L139 should H2O have a sub-scripted 2

> This was corrected in the revised manuscript.

Comment 3: Were the differences in sampling lines of the WIBS and UV-APS calculated? Reynolds number for instance?

> We did not calculate the Reynolds number or quantify possible difference in the two sampling lines. The lines from the inlets were somewhat different in length (~4.5 m for the UV-APS and <1 m for the WIBS), but both were arranged to minimize bends and were oriented vertically. Thus, the differences in particle number concentration from the inlets and lines is likely minimal.

Comment 4: Were all particles assumed to be spherical for the density calculations or was the WIBS ability to determine shape utilized?

> All particles were assumed to be spherical for particle mass calculations. Particle morphology could impact particle mass calculations, however, the asymmetry factor (AF) provided by the WIBS has not been characterized sufficiently to understand the relationship of this parameter to particle morphology. As a result, we did not utilize AF. To clarify ambiguity, the text was revised at L160-162 as follows:
> "Total particle number size distributions (irrespective of fluorescence properties) obtained from the UV-APS *and WIBS* were converted to mass distributions  *assuming spherical particles of* unit particle mass density as a first approximation , unless otherwise stated."

Comment 5: Do you believe that cluster 1 is solely a fungal spore cluster, given its size range overlaps with that of some bacteria?

The organization of clusters from the raw data is a function of the mathematical algorithms utilized and is relatively robust. The assignment of names or sources to the derived clusters is much more uncertain. While Crawford et al. (2015) assigned Cluster 1 to be fungal spores, this should be taken loosely. It is very possible that some fraction of non-fungal particles have been conflated with this cluster. Without direct comparative evidence there is no way to confidently know the source or category of each particle. For example, even the cluster assignment of even polystyrene latex particles of known type was reported as only 98% in a previous publication (Crawford et al., 2015). To clarify this point we have added the following text to the end of L203. "*It should be noted that assignment of names and approximate origin (e.g. fungal spores or bacteria) to clusters is approximate and does not imply particle homogeneity. Each cluster likely contains a small percentage of contaminating particles. For more details see Robinson et al. (2013) and Crawford et al. (2015).*"

Comment 6: Why was a rainfall accumulation threshold of greater than 0.201 chosen?

A threshold of 0.201 represents a normalized and unitless value that takes into account both disdrometer and tipping bucket measurements. This value was chosen arbitrarily based on the following reasoning. Rain events that presented <0.201 often did not coincide with other indicators of rain such as increased fluorescent particle concentration and RH. When the threshold value was increased to 0.201 we observed more continuity in the measurements that are indicative of rain events.

Comment 7: What did the correlations look like before the manual reclassification of some of the rain/dry periods? How much did this effect it?

Regarding the correlations, manual reclassification by wetness category increased the $R^2$ values in all cases. For example, prior to reclassification the mass correlation of arabitol with WIBS cluster 1 during rainy periods was 0.77 after reclassification the $R^2$ value was 0.82. This trend of increased $R^2$ was observed with other correlations for both rainy and dry periods.

Comment 8: L 430-431. The Hill 2009 reference does talk about increased wetness effecting the fluorescent properties in comparison to dry samples however in this study wet samples were particles suspended in solution rather than particles at higher relative humidity's. **(a)** I believe that this line should be rewritten. **(b)** Do you believe the particles sampled during wet periods to be in droplets or to have increased moisture content? **(c)** Could a moistened PBAP have increased fluorescence due to fluorescent compounds being extracted/leached to its surface?

These are interesting questions that were somewhat beyond the scope of the ambient study performed here and thus we did not fully investigate them.
**(a)** Taking this comment into account we revised this sentence (L 430-431) to be more accurate: "This *could* impact the fluorescence properties of the fungal spore particles *that have different amounts of adsorbed or associated water* (Hill et al., 2009; 2013; 2015)."
**(b)** As far as the moisture content of individual spores, we have no direct evidence either way. It is possible that some of the spores were fully contained within water droplets, either as a by-product of the high RH and deliquescence or because spores were actively ejected by fungus and thus encased in a small droplet. Upon interrogation within the UV-LIF instruments, however, the spores were almost surely not activated within a droplet, because of the size ranges observed. If they were encased within a droplet the average size would have likely been too large for the UV-

LIF instruments to sample efficiently and we would not have observed the dominant 2-6 μm modes.

**(c)** We are aware of no studies that directly link increased fluorescence with the leaching of fluorescent compounds from the interior to the surface of a particle. However, (Hill et al., 2013; 2015) showed that the water content associated with bacterial aerosols significantly affected their fluorescence properties, which led to the brief statement quoted above.

Comment 9: Was there much difference in fluorescent intensity for FAP on Dry and Wet periods?

We did not perform this analysis as a part of this study. But, intrigued by the referee's question we calculated average fluorescence intensity from two samples (one Rainy, one Dry) as examples. Hi Vol sample 8 was a dry sample with intensities as follows: FL 1, 872 ± 718; FL 2, 654 ± 277; FL 3, 497 ± 347. Hi Vol sample 16 was a rainy sample with intensities as follows: FL 1, 1687 ± 613; FL 2, 740 ± 333; FL 3, 707 ± 493. In this example, FL1 intensity increased by a factor of 2, FL2 intensity only nominally increased, and FL3 intensity increased by ~40%.

Comment 10:
- L555 Should "Figures 6 c-f" read "Figures 6 d-f"?
  - Corrected
- L560 Should "Figure 6 c, d" read "Figure 6 d, e"?
  - Corrected
- L567 Should "Figure 6 e, f" read "Figure 6 g, h"?
  - Corrected

Comment 11: For the total particulate matter mass concentrations why did you not use the high volume sampler samples to determine the total mass? Instead of the UV-APS measurements.

- Filter mass was not measured before and after sampling and so it was not possible to estimate total particle mass using these filters. As a result, we estimated particle mass using the integrated mass from a particle sizing instrument.

Comment 12: You mention Cladosporium are generally present/released at dry periods was there any evidence that this occurred during this campaign?

The observation that *Cladosporium* spores are present in highest concentration during dry periods has been reported many times and is generally well accepted (De Groot, 1968; Oliveira et al., 2009). For example, it was shown for a study in rural Ireland that both WIBS and UV-APS instruments poorly detected *Cladosporium* particles (Healy et al., 2014). Unfortunately we have no direct observations of this from the campaign. We collected particle by impaction (Sporewatch drum sampler), but it malfunctioned and we have no direct microscopy samples to show relative spore concentrations. The DNA analysis shows relative diversity, but does not provide quantitative evidence that can support the suggestion that *Cladosporium* was present primarily during dry periods.

**References**

Crawford, I., Ruske, S., Topping, D., and Gallagher, M.: Evaluation of hierarchical agglomerative cluster analysis methods for discrimination of primary biological aerosol, Atmos Meas Tech, 8, 4979-4991, 2015.

De Groot, R.: Diurnal cycles of air-borne spores produced by forest fungi, Phytopathology, 58, 1223-1229, 1968.

Healy, D., Huffman, J., O'Connor, D., Pöhlker, C., Pöschl, U., and Sodeau, J.: Ambient measurements of biological aerosol particles near Killarney, Ireland: a comparison between real-time fluorescence and microscopy techniques, Atmos Chem Phys, 14, 8055-8069, 2014.

Hill, S. C., Mayo, M. W., and Chang, R. K.: Fluorescence of bacteria, pollens, and naturally occurring airborne particles: excitation/emission spectra, DTIC Document, 2009.

Hill, S. C., Pan, Y.-L., Williamson, C., Santarpia, J. L., and Hill, H. H.: Fluorescence of bioaerosols: mathematical model including primary fluorescing and absorbing molecules in bacteria, Optics Express, 21, 22285-22313, 10.1364/oe.21.022285, 2013.

Hill, S. C., Williamson, C. C., Doughty, D. C., Pan, Y.-L., Santarpia, J. L., and Hill, H. H.: Size-dependent fluorescence of bioaerosols: Mathematical model using fluorescing and absorbing molecules in bacteria, Journal of Quantitative Spectroscopy and Radiative Transfer, 157, 54-70, http://dx.doi.org/10.1016/j.jqsrt.2015.01.011, 2015.

Oliveira, M., Ribeiro, H., Delgado, J., and Abreu, I.: The effects of meteorological factors on airborne fungal spore concentration in two areas differing in urbanisation level, International journal of biometeorology, 53, 61-73, 2009.

---

## Author Comment (AC2) · 4 Nov 2016

Note regarding document formatting: black text shows original referee comment, blue text shows author response, and red text shows quoted manuscript text. Changes to manuscript text are shown as *italicized and underlined*. All line numbers refer to discussion/review manuscript.

General Comments: The article is of high quality providing a novel information relevant to the ACP addressing the atmospheric biological (fungal) tracers. The novelty is in correlations reported for periods affected by rain between fungal biomarkers obtained from offline measurements and fluorescent aerosol particle concentrations obtained by direct online measurements. The description of experimental work is sound and detailed supporting the good quality of the paper.

Author response: We thank the referee for his/her positive assessment and summary.

Specific Comments:
(Note that referee comments have been labeled by number and chopped by individual referee-thought so they can be dealt with in a clear sequence)

Comment 1: In my opinion, the article would gain if additional data with regard to total PM mass concentrations were reported. For example Table 3 presents % contribution of biomarkers with regard to particulate matter and spore mass. The estimated PM mass data presented along with the rest of the data would help to clarify relationship to overall chemical characterization of PM if The data reported are comprehensive.

Total particle mass ($\mu g\ m^{-3}$) was added to Table S4.

Comment 2: Still are there also data available for the same period reporting on the occurrence of organic carbon and thus allowing for discussion of traditionally reported chemical characterization of organic particulate matter?

Total organic carbon measurements for the same sampling periods are not available. We asked several BEACHON-RoMBAS collaborators, but did not find such data available.

Comment 3: Authors report on taxonomic differences in fungal DNA during wet and dry periods. Could such differences be attributed to the ability of different fungal species to survive in different humidity conditions?

It is certainly plausible that certain fungal species are more likely to survive in wet conditions, or vice versa, and that the rate of emission of a given species will be lower during conditions unfavorable for survivability. However, unless the DNA were to become damaged, which is unlikely, the molecular genomic analyses will still detect the presence of the species. So this process could be involved on a small level, but it is unlikely that survivability would directly impact the observations.

---

## Author Comment (AC3) · 4 Nov 2016

Note regarding document formatting: black text shows original referee comment, blue text shows author response, and red text shows quoted manuscript text. Changes to manuscript text are shown as italicized and underlined. All line numbers refer to discussion/review manuscript.

General Comments: The manuscript entitled "Fluorescent Bioaerosol Particle, Molecular Tracer, and Fungal Spore Concentrations during Dry and Rainy Periods in a SemiArid Forest" by Gosselin et al. reports correlations of fluorescent aerosol particles of UV-APS and WIBS-3 with molecular tracers of fungal spores and bacteria. This study provides further investigations of the detection ability of UV-LIF instruments of fungal spores. In general, the manuscript was well written and the analysis of the data was well performed. I recommend this manuscript to be accepted for publication after minor revisions.

Author response: We thank the referee for his/her positive assessment and summary.

Specific Comments:
Comment 1: In the last paragraph of Introduction and the Discussion sections, the authors declared that this is the first comparison of online UV-LIF with organic molecular tracers measurements. In fact, a recent study has also made such comparisons between WIBS and fungal spore tracers (see Yue et al., 2016, Sci. Rep.).

We thank the referee for pointing out this reference that we have now included at L127. The Yue et al. paper indeed briefly presents arabitol and mannitol concentrations and also shows WIBS data during one rain event, but does so by showing only qualitative relationships between WIBS and tracer measurements without presenting any quantitative correlations. We have edited the text at L132 to the following to be more accurate with respect to the inclusion of the Yue et al. reference:
"This study of ambient aerosol represents the first quantitative comparison of real-time aerosol UV-LIF instruments with molecular tracers or culturing."

The Yue paper is also discussed and references in the text at L480:
"More recently, Yue et al. (2016) studied a rain event in Beijing and observed increased polyol concentrations at the onset of the rain. The observed mannitol concentration (45 ng m-3) was approximately consistent with observations reported here and with previous reports, while the arabitol concentration values observed were approximately an order of magnitude lower (0.3 ng m-3)."

Comment 2: In part 2.2 Online fluorescent instruments (Line 174 – 176), the fluorescent detection bands for WIBS-3 should be λem 310 – 400 nm and λem 400 – 600 nm (see Gabey et al., 2010, ACP). Please clarify it.

The WIBS-3 was not a commercialized instrument and so different models had slightly different detector properties. Crawford et al. (2014) reports the following parameter for the PMT detectors: "excitation wavelengths centred at 280±10 nm and 370±20 nm" and emission in "one of two bands that do not overlap the excitation emission, 320–400 nm and 410–650 nm." We have

adjusted the lower bound of the FL1 emission channel from 310 nm to 320 nm to match the Crawford et al. values (L175-176).

Comment 3: Line 205: Provide references for "One important difference between the models is that the WIBS-3 exhibits comparatively weak FL1 and FL2 signals with respect to the more updated models, and is thus more influenced by FL3".

We have clarified the text after L205:
"One important difference between the models is that the  *optical chamber design and filters of the WIBS-4 models were updated to enhance the overall sensitivity of the instrument (Crawford et al., 2014). Additionally, slight differences in detector gain between models and individual units can impact the relative sensitivity of the fluorescence channels. . This may result* in differences in fluorescent channel intensity between instrument models, as will be discussed later."

Comment 4: In Figure 5 (e, f), the unit for WIBS Cl1 FAP was given as mass concentration. How do the authors convert the number concentrations to mass concentrations for WIBS-3? Such information should be provided in the Methods section.

For all mass concentration data reported in the manuscript we took UV-APS or WIBS-3 number size distributions, assuming spherical particles with unit density, and converted to mass distributions (mass = number x 4/3 pi x r^2), where r is the particle diameter. Integrated mass concentrations were calculated by integrating the total mass between 0.5 and 15 µm. This process is detailed in the discussion version of the paper at L159-167, but has been revised slightly as detailed below:
"Total particle number size distributions (irrespective of fluorescence properties) obtained from the UV-APS *and WIBS* were converted to mass distributions  *assuming spherical particles of* unit particle mass density as a first approximation , unless otherwise stated."

**References**

Crawford, I., Robinson, N. H., Flynn, M. J., Foot, V. E., Gallagher, M. W., Huffman, J. A., Stanley, W. R., and Kaye, P. H.: Characterisation of bioaerosol emissions from a Colorado pine forest: results from the BEACHON-RoMBAS experiment, Atmos Chem Phys, 14, 8559-8578, 10.5194/acp-14-8559-2014, 2014.